# Whole-genome sequencing reveals the molecular implications of the stepwise progression of lung adenocarcinoma

Yasuhiko Haga[1,12], Yoshitaka Sakamoto[1,12], Keiko Kajiya[1,12], Hitomi Kawai [2], Miho Oka[1,3], Noriko Motoi [4,5], Masayuki Shirasawa[6,7], Masaya Yotsukura[8], Shun-Ichi Watanabe[8], Miyuki Arai[1], Junko Zenkoh[1], Kouya Shiraishi [7,9], Masahide Seki [1], Akinori Kanai [1], Yuichi Shiraishi [10], Yasushi Yatabe [4], Daisuke Matsubara[2], Yutaka Suzuki [1] ✉, Masayuki Noguchi[2,11], Takashi Kohno [7] ✉ & Ayako Suzuki [1] ✉

The mechanism underlying the development of tumors, particularly at early stages, still remains mostly elusive. Here, we report whole-genome long and short read sequencing analysis of 76 lung cancers, focusing on very early-stage lung adenocarcinomas such as adenocarcinoma in situ (AIS) and minimally invasive adenocarcinoma. The obtained data is further integrated with bulk and spatial transcriptomic data and epigenomic data. These analyses reveal key events in lung carcinogenesis. Minimal somatic mutations in pivotal driver mutations and essential proliferative factors are the only detectable somatic mutations in the very early-stage of AIS. These initial events are followed by copy number changes and global DNA hypomethylation. Particularly, drastic changes are initiated at the later AIS stage, i.e., in Noguchi type B tumors, wherein cancer cells are exposed to the surrounding microenvironment. This study sheds light on the pathogenesis of lung adenocarcinoma from integrated pathological and molecular viewpoints.

Lung cancer is a major cause of death worldwide. An increasing number of patients are diagnosed with lung adenocarcinoma every year, rendering it the most frequently detected subtype of non-small cell lung cancer (NSCLC). In 1995, Noguchi et al. reported the "non-invasive" adenocarcinomas of the lung. They are small-sized adenocarcinomas (≤2 cm in diameter) that have classified into two groups[1,2]: replacement and non-replacement adenocarcinomas. The former was further sub-classified into three subtypes (types A, B, and C). Patients with types A and B adenocarcinomas (pure-lepidic adenocarcinomas) have highly favorable prognosis and are considered very-early-stage adenocarcinomas, known as adenocarcinoma in situ (AIS). Thereafter, the prognosis of these patients rapidly deteriorates, with the 5-year survival rate

[1]Department of Computational Biology and Medical Sciences, Graduate School of Frontier Sciences, The University of Tokyo, 5-1-5 Kashiwanoha, Kashiwa, Chiba 277-8561, Japan. [2]Department of Diagnostic Pathology, Faculty of Medicine, University of Tsukuba, 1-1-1 Tennodai, Tsukuba, Ibaraki 305-8575, Japan. [3]Ono Pharmaceutical Co., Ltd., Ibaraki, Japan. [4]Department of Pathology, National Cancer Center Hospital, 5-1-1 Tsukiji, Chuo-ku, Tokyo 104-0045, Japan. [5]Department of Pathology, Saitama Cancer Center, 780 Komuro, Ina, Kita-Adachi-gun, Saitama 362-0806, Japan. [6]Department of Thoracic Oncology, National Cancer Center Hospital, 5-1-1 Tsukiji, Chuo-ku, Tokyo 104-0045, Japan. [7]Division of Genome Biology, National Cancer Center Research Institute, 5-1-1 Tsukiji, Chuo-ku, Tokyo 104-0045, Japan. [8]Department of Thoracic Surgery, National Cancer Center Hospital, 5-1-1 Tsukiji, Chuo-ku, Tokyo 104-0045, Japan. [9]Department of Clinical Genomics, National Cancer Center Research Institute, 5-1-1 Tsukiji, Chuo-ku, Tokyo 104-0045, Japan. [10]Division of Genome Analysis Platform Development, National Cancer Center Research Institute, 5-1-1 Tsukiji, Chuo-ku, Tokyo 104-0045, Japan. [11]Clinical Cancer Research Division, Shonan Research Institute of Innovative Medicine, Shonan Kamakura General Hospital, 1370-1 Okamoto, Kamakura, Kanagawa 247-8533, Japan. [12]These authors contributed equally: Yasuhiko Haga, Yoshitaka Sakamoto, Keiko Kajiya. ✉e-mail: ysuzuki@edu.k.u-tokyo.ac.jp; tkkohno@ncc.go.jp; asuzuki@edu.k.u-tokyo.ac.jp

of type C adenocarcinoma with fibroblastic proliferation foci being 75%. In the current World Health Organization (WHO) histological classification[3], lung adenocarcinomas (≤3 cm in diameter) showing pure-lepidic growth are classified collectively as AIS. AIS corresponds to Noguchi types A and B to some extent and show a 100% 5-year patient survival rate. However, to ensure the accurate diagnosis of AIS, WHO has introduced a new type of adenocarcinoma called minimally invasive adenocarcinoma (MIA). These adenocarcinomas have a small invasion area (≤5 mm) and a likely have an extremely favorable prognosis.

In both pathological criterion, AIS is defined as a pure-lepidic, patterned tumor with a diameter of ≤3 cm that shows no relevant invasion. AIS shows lepidic growth, mimicking non-tumorous lung alveoli. It does not exhibit invasive features, such as having non-lepidic histological subtypes, causing necrosis, and leading to vascular and lymphatic invasions. MIA also has a diameter of ≤3 cm, but it is characterized by its small invasion area (≤5 mm). It is considered to eventually progress to the overtly invasive adenocarcinoma (≥3 cm). Tumor can also develop into more malignant and distinct histological subtypes, such as papillary, acinar, micropapillary, and solid subtypes, other than the lepidic subtype. Although patients with an invasive lung adenocarcinoma shows a poor prognosis, those with AIS and MIA can be successfully treated by performing surgical resection[4]. Therefore, to understand additional and essential molecular events that lead to the development of invasive adenocarcinoma, it is crucial to characterize its molecular progression from being a pre-invasive (AIS or MIA) to overtly invasive adenocarcinoma.

Recently, several researchers worldwide, The Cancer Genome Atlas[5,6], and International Cancer Genome Consortium (ICGC)[7,8] have attempted to sequence the lung adenocarcinoma genome. In particular, the Pan-Cancer Analysis of Whole Genomes[9] consortium of ICGC conducted the whole-genome sequencing (WGS) analysis of lung adenocarcinomas, covering single nucleotide variants (SNVs) and large-scale structural variants (SVs). However, early-stage cancers are not sufficiently represented in these datasets, as the analyzed cases mainly include cancers of advanced stages with invasive characteristics. Recently, some studies have reported genomic and epigenomic features and their heterogeneity in early lung adenocarcinoma genomes[10–12]. However, these studies only performed short read sequencing in limited regions of the genome (i.e., the exome) or focused on AIS parts within overall advanced tumors. Thus, a precise and comprehensive characterization of SVs and other complicated genomic aberrations has not been achieved so far. Notably, long read sequencing could not be performed for lung adenocarcinoma genomes until very recently, as the meager amount of required starting materials hampered intensive analyses of small-tumor specimens. Our group recently reported a long read WGS analysis on a limited number of NSCLCs. We elucidated complicated SVs[13] and chromothripsis patterns in a haplotype-aware manner[14].

In this study, we scale-up our short and long read WGS analysis, while particularly focusing on early lung adenocarcinomas. A total of 76 lung cancers, including AIS (Noguchi type A and B), MIA and lepidic adenocarcinoma cases, are subjected to long and short read whole genome analysis. Each case is subjected to a close association study with pathological classifications. We further integrate these results with bulk and spatial transcriptomic and epigenomic analyses of the same specimens. Here, we describe a comprehensive integrated multi-omics analysis of early-stage lung adenocarcinoma at representative pathological stages.

## Results

### Whole-genome sequencing (WGS) analysis of early and advanced adenocarcinomas

Whole-genome short read and long read sequencing datasets of 76 lung cancer specimens were analyzed. The datasets included newly generated data for 48 early small-sized lung adenocarcinoma cases (collectively called "Early-Ad" hereafter). These cases included 26 AIS (9 and 17 cases of Noguchi type A and type B, respectively), 18 MIA, and 4 lepidic adenocarcinoma ("Lepidic-Ad"; representative histological images are shown in Fig. 1a) cases. Additionally, 8 advanced adenocarcinoma (called "Advanced-Ad" hereafter) cases were also analyzed, and their data were integrated with data of 20 additional Advanced-Ad and other NSCLC cases for which we generated WGS data in our previous studies[13] (Fig. 1b). Detailed clinical and pathological information is presented in Table 1 and Supplementary Table S1. The overall analytical procedure is also shown in Supplementary Fig. S1.

For newly generated datasets of early lung cancers, we obtained an average of 1,086,182,079 and 689,436,667 short read paired-end sequencing pairs, corresponding to 97× and 62× average sequencing depths for tumor and paired normal samples, respectively (Supplementary Table S2). Long read sequencing analysis was successfully performed for 37 Early-Ad cases using PromethION (Oxford Nanopore Technologies [ONT]) at an average sequencing depth of 24× and 28× (73 and 82 Gb), respectively. N50 read lengths of 24 kb and 22 kb were used for tumor and normal samples, respectively (Supplementary Table S3). WGS data from 25 invasive NSCLCs with Advanced-Ad and various pathological subtypes was partly obtained in our previous study[13] (Supplementary Tables S2 and S3).

Generally, short read sequencing data was used to detect somatic mutations, including point mutations, SVs, and copy number variants (CNVs). Long read sequencing data was used to precisely detect larger and more complex SVs to profile DNA methylation patterns and perform haplotype-aware analyses. Details of each procedure are mentioned in the Methods section.

### Somatic mutation patterns of the driver and other key genes

The analysis of obtained datasets revealed well-known *EGFR* driver mutations, such as L858R (16 cases), exon 19 deletion (9 cases), exon 20 insertion (3 cases), and other hotspots (2 cases), in 30 (63%) Early-Ad cases. These somatic mutations are thus already acquired in the early stages of adenocarcinomas as the first aberrant event we could detect (Fig. 1b). The frequencies of *EGFR* mutations were similar in Early-Ad cases of different pathological classes and advanced cases (AIS Noguchi type A: 56%, type B: 71%, MIA and Lepidic-Ad: 59%, Advanced-Ad cases: 65%) (Fig. 1c). Previous studies have partly indicated the same results[10,11,15]. In other Early-Ad cases, *MET* (Y1021N, exon 14 splice site, exon 14 deletion), *KRAS* (G12A, G12D)[16], *BRAF* (A489_Q493del)[17], *ERBB2* (V659E)[18], and *MAP2K1* (E102_I103del)[19] mutations, and *RET* (KIF5B-RET) and *ALK* (EML4-ALK) fusions were detected as other driver aberrations[20–23]. Eight cases (17%) were identified as pan-negative tumors. The observed driver compositions were broadly similar to those observed in advanced adenocarcinomas, suggesting that most important key driver mutations occur at early stages.

Contrary to the driver mutations, only seven (15%) Early-Ad cases harbored relevant somatic mutations in tumor-suppressor genes. In the Advanced-Ad cases, deleterious mutations in tumor-suppressor genes were more frequent (48%, 11 cases; $p = 0.0072$, Fisher's exact test) (Fig. 1b). These results indicated that driver mutations are already acquired in most AIS cases; however, somatic mutations in tumor-suppressor genes, particularly in *TP53* ($p = 0.0012$, Fisher's exact test) were not observed in these cases (Fig. 1c).

We also profiled CNVs using the short read data. Oncogenic amplifications[20,24], such as increase in the copy-number of the *MYC* and *TERT* regions, were mostly observed in advanced cases (Fig. 1d). Especially in tumor-suppressor gene regions, such as at *TP53*, copy number loss events were almost exclusively observed in advanced cases. A small number of AIS cases harbored some copy number aberrations in the proximal genomic regions of cancer-related genes (Fig. 1e). In accordance with previous reports[2,25], gene deletion and/or copy number loss events of *CDKN2A* were identified in several cases;

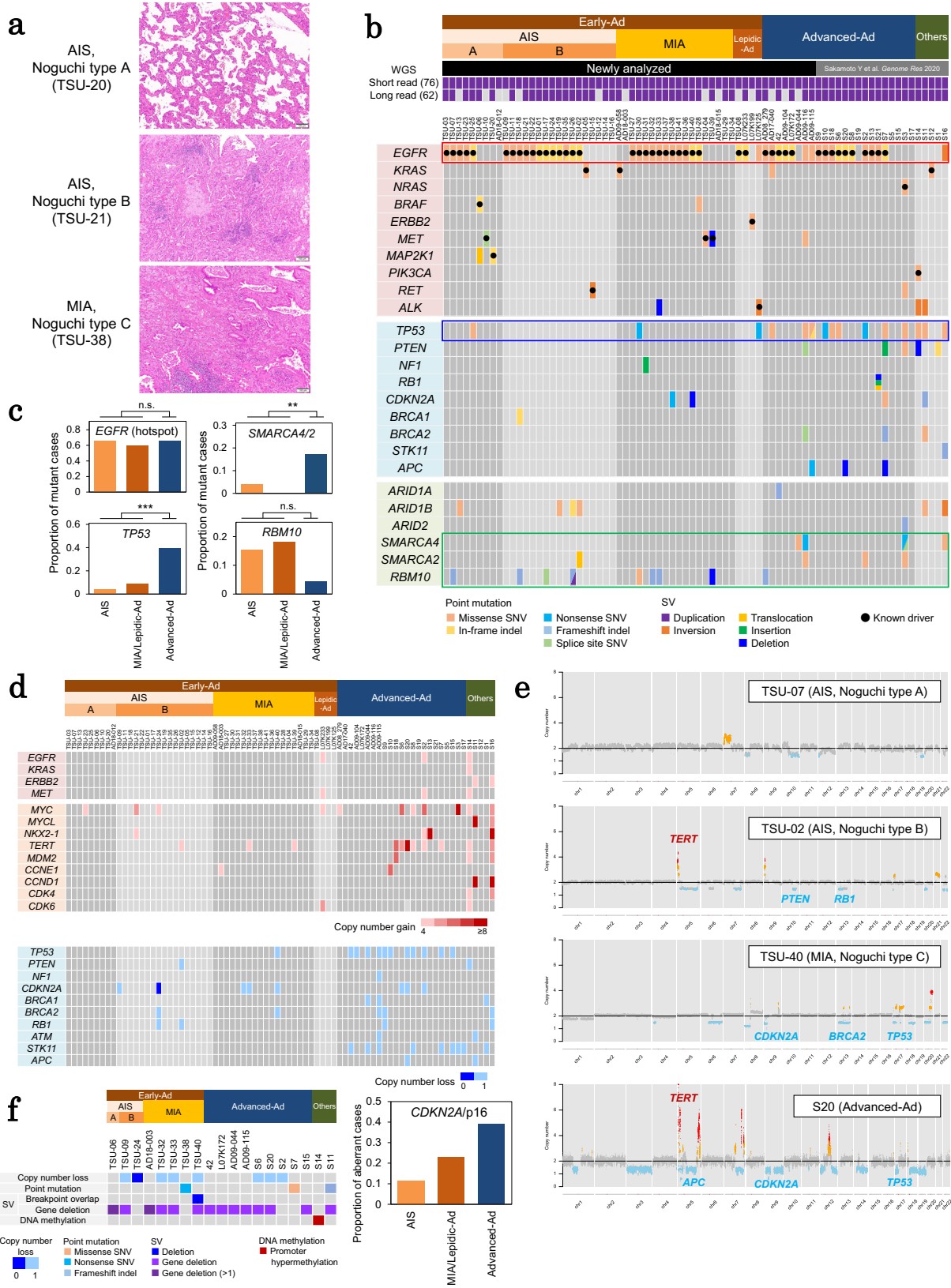

they were also observed in Early-Ad stages, although the disruption of tumor-suppressor genes rarely occur at these stages (Fig. 1f and Supplementary Fig. S2). These results suggested that key driver gene mutations initiate the development of early stages of cancer, followed by the deletion of this gene, which might promote cancer cell proliferation during early progression of adenocarcinomas.

In addition to pivotal driver and tumor-suppressor mutations, several mutations were detected in some cancer genes associated with chromatin remodeling and RNA splicing, including *SMARCA4* and *SMARCA2* (SWI/SNF chromatin-remodeling factor genes). These somatic mutations were also mostly detected in advanced cases (p = 0.035, Fisher's exact test) (Fig. 1b, c and Supplementary Fig. S2).

**Fig. 1 | Basic characterization of cancer-related genes from AIS to advanced lung adenocarcinomas. a** Representative H&E images of Early-Ad cases with Noguchi classification. We analyzed 9, 19, 20 of Noguchi type A, B and C Early-Ad cases, respectively, in this study. Images show representative examples of tissue regions for each histological subtype, which were annotated by the pathologists. **b** The somatic mutation status of cancer-related genes. Point mutation and SV statuses of oncogenes, tumor-suppressor genes and other known mutated genes are shown in each case. **c** The proportion of cases with *EGFR* hotspot mutations (e.g., L858R, exon 18/19 deletions and exon 20 insertions), *TP53* mutations, *SMARCA4* and *SMARCA2* mutations, and *RBM10* mutations in the corresponding AIS, MIA/Lepidic-Ad and Advanced-Ad cases. When comparing Early-Ad and Advanced -Ad cases, the *p*-values are indicated by asterisks, **\*\*p < 0.05, \*\*\*p < 0.005.**

n.s.: not significant (*p* ≥ 0.05). The *p*-values were calculated by Fisher's exact test (two-sided, no multiple comparison adjustments). **d** CNV status of cancer-related genes. Copy number gains for the representative oncogenes and genes frequently reported with amplifications in lung cancers (upper panel). Copy number losses are also shown for the representative tumor suppressor genes (lower panel). **e** Copy number profiles of representative cases. Cancer-related genes affected by copy number aberrations are shown in the inset. **f** Aberrant events in *CDKN2A* region in each case (left). Proportion of cases with *CDKN2A* aberrations in the corresponding AIS, MIA/Lepidic-Ad, and Advanced-Ad cases (right). CN copy number. Whole-gene deletion (>1): whole-gene deletions detected with ≥2 deletion breakpoint pairs, which possibly indicates homozygous deletions. Source data are provided as a Source Data file for (**b**).

## Table 1 | Clinical and pathological information of early lung adenocarcinoma cases

| Early/advanced | Classification | Total | Age (avg.) [Range] | Sex | | Smoking history | |
|---|---|---|---|---|---|---|---|
| | | | | Female | Male | y | n |
| Early | AIS, Noguchi type A | 9 | 70 [62–75] | 4 | 5 | 4 | 5 |
| Early | AIS, Noguchi type B | 17 | 72 [50–87] | 10 | 7 | 8 | 9 |
| Early | MIA | 18 | 68 [64–75] | 14 | 4 | 7 | 11 |
| Early | Lepidic-Ad | 4 | 59 [50–65] | 2 | 2 | 2 | 2 |
| Advanced | Advanced-Ad | 23[a] | 63 [38–85] | 12 | 11 | 10 | 13 |
| Advanced | Others | 5[a] | 67 [50–85] | 1 | 4 | 5 | 0 |

[a]The sequencing data of 15 Advanced-Ad and 5 Others were obtained and analyzed in our previous study[13,14].

Interestingly, a loss-of-function mutation in the *RBM10* gene, considered to be involved in RNA splicing[10,26], was also detected in nine cases. Unlike other somatic mutations, these mutations were commonly observed in Early-Ad cases (*p* = 0.25, Fisher's exact test; the *p*-value was not statistically significant due to the small total number of mutated cases). Previous reports indicated that aberrations in the function of this gene can cause aberrant splicing events. Liu et al. reported that *RBM10* deficiency can induce immune cell infiltration by producing neoantigens via aberrant splicing[26]. Despite the possible advantages of *RBM10* mutations in the early stage of cancer initiation, they may have a deleterious effect in later stages, i.e., when the cancer spreads and is attacked by immune cells (see the section below on transcriptional changes during cancer development).

It can be inferred that these mutational profiles indicate the acquisition of driver mutations and loss of *CDKN2A*/p16 are the first and major events in very early AIS cases, with other genomic events occurring at later stages.

**Long read sequencing analysis for precise detection of large SVs**
Using long read data (Supplementary Table S3), we detected SV using Nanomonsv[27]. An average of 153 SVs with varying lengths were detected (Fig. 2a), including an oncogenic driver fusion *KIF5B-RET* and *MET* exon 14 deletion (Fig. 2b). In advanced cases, breakpoints of deleterious SVs were detected in tumor suppressor genes, such as *APC* and *RB1*. As previously reported, these SVs were not detected in Early-Ad cases.

We also detected SVs with short read data (Supplementary Fig. S3a). Overall, 57% SVs from the short read data could be directly verified with ≥1 long read(s) (Fig. 2c). The rate of possible false-positive detections of short reads, which varied among SV types, is shown in Supplementary Fig. S3b. Translocations detected using short read WGS were relatively less reliable, as few translocations were covered by long read WGS compared with other SVs.

Then, we scrutinized SVs associated with known repetitive sequences and mobile elements. An average of 16 (0–243) inserted sequences were detected (Fig. 2d and Supplementary Fig. S3c). Of these, an average of 6 (41%) inserted sequences were classified as retrotransposons, including LINE-1 (solo L1), L1 transductions, Alu, SINE-VNTR-Alu (SVA), and processed pseudogenes (PSD). A substantial number of LINE-1 insertions were detected in the squamous cell lung carcinoma case S14, which aligns with previous reports indicating that this phenomenon commonly occurs in squamous cell lung carcinomas[28]. Previous studies reported that the activation of retrotransposable elements, especially LINE-1, may promote rearrangements of cancer genomes. Moreover, the association of transpositions with the degree of DNA hypomethylation has also been reported[28–30].

**Mutation abundance**
We analyzed changes in somatic mutation abundances (Fig. 3 and Supplementary Fig. S4). An average of 1.9 somatic point mutations was detected per Mb in the Early-Ad cases. Surprisingly, approximately same number of somatic mutation were observed in Advanced-Ad cases (2.6 mutations per Mb), except for six "hyper-mutator" cases (11–158 mutations/Mb). Similar results were also obtained for SVs, indicating that the number of somatic mutation events should not significantly differ depending on the tumor stage or histological class, unless a "hyper-mutation" event occurs (Fig. 3a). Moreover, "hyper-mutation" was detected only among Advanced-Ad cases or cases of other NSCLC subtypes; therefore, AIS cases were not affected by this issue (Fig. 3a). We also found that some CNVs already occurred in AIS. Cases associated with several CNV events appeared in the MIA/Lepidic-Ad and later stages (Fig. 3a and Supplementary Fig. S4b).

Interestingly, we found that *EGFR* mutant cases harbored a high number of genomic alterations at AIS stages (Fig. 3b and Supplementary Fig. S4c). Conversely, this difference was not detected in Advanced-Ad cases (Fig. 3b). EGFR interacts with DNA repair molecules, such as DNA-PK and ATM, and activates double-strand break

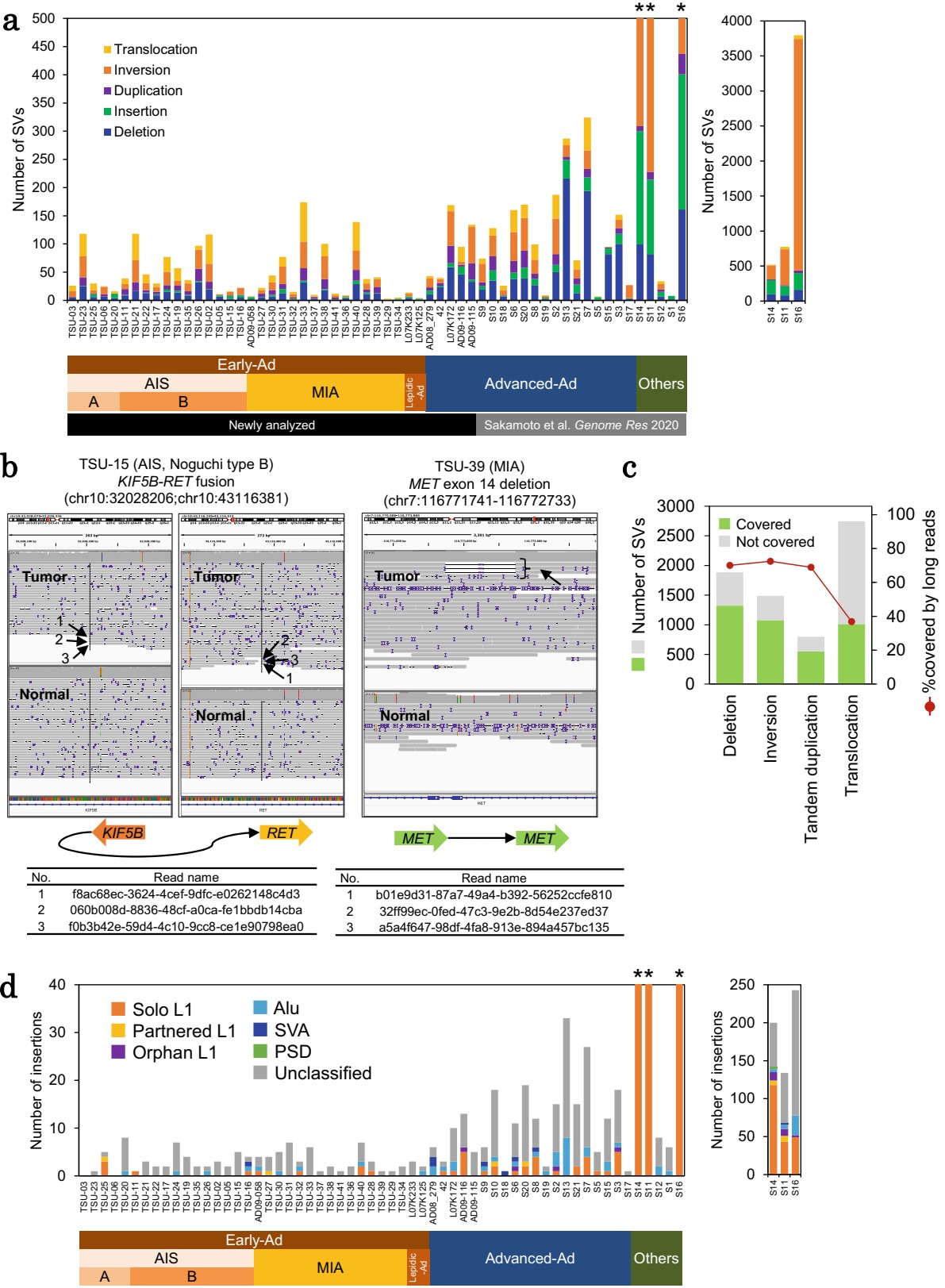

**Fig. 2 | Precise detection of somatic SVs by long read sequencing. a** Number of somatic SVs detected in long read WGS using Nanomonsv. The number of SVs for the three cases with the highest numbers of SVs (asterisk) is also shown in the separate graph. **b** An example of SVs in Early-Ad cases. Driver events (*KIF5B-RET* fusion and *MET* exon 14 deletion) for the corresponding cases. Both primary and supplementary alignments visualized by IGV. The truncated read names are provided in the bottom tables. **c** Long read coverage of SV junctions detected with short read WGS. **d** Number of insertions detected from long read WGS using Nanomonsv. Number of inserted sequences for three cases with the highest number of insertions (asterisk) is also shown in the separate graph. Source data are provided as a Source Data file for (**a**, **b** and **c**).

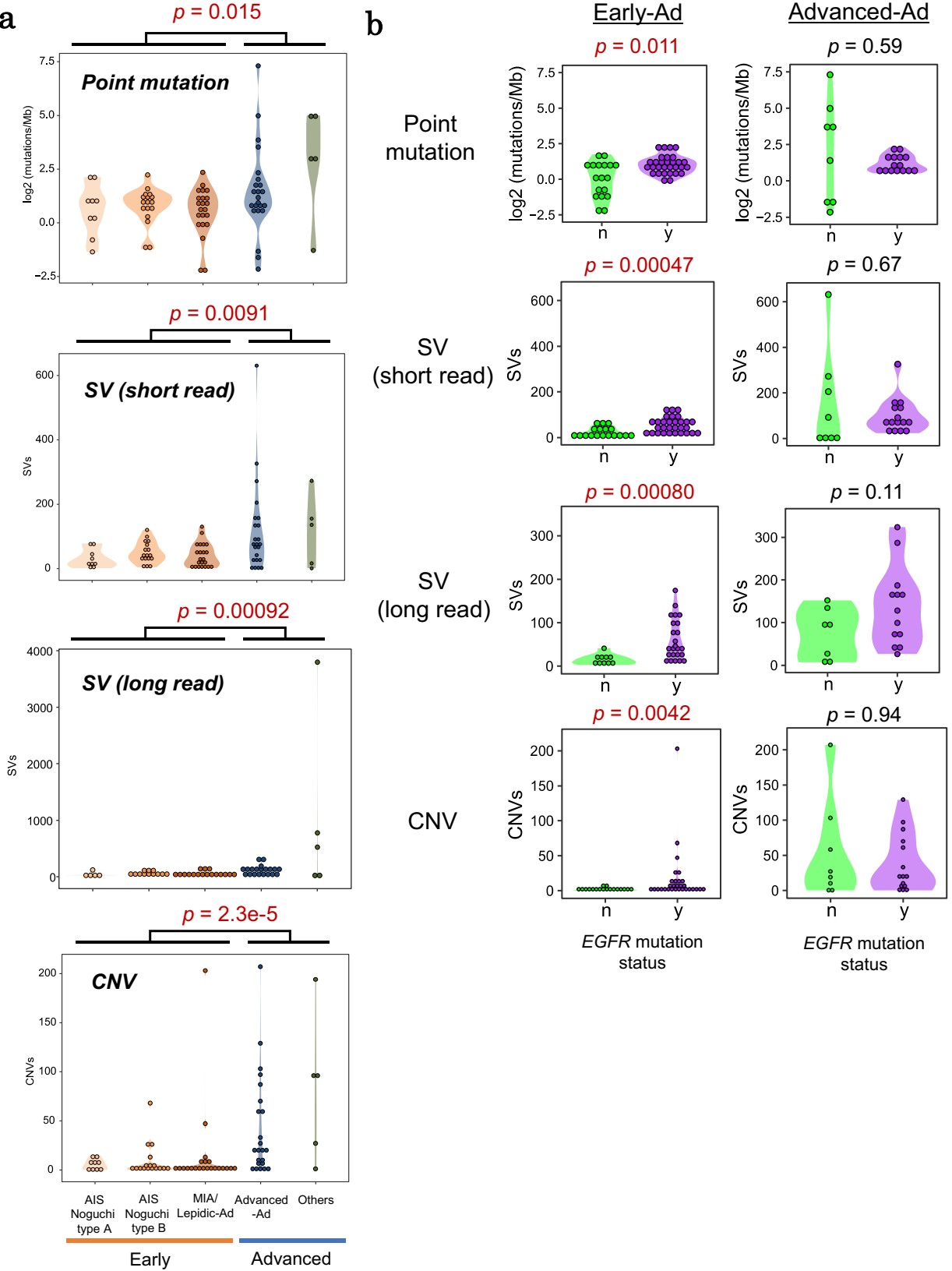

**Fig. 3 | Patterns of somatic mutation accumulations. a** Somatic mutation abundances in the corresponding groups. The number of point mutations, SVs, and CNVs is shown in the upper, middle, and lower panels, respectively. The *p*-values were calculated by Wilcoxon rank sum test (two-sided, no multiple comparison adjustments). **b** Comparison of the number of somatic mutations between *EGFR* mutation−positive and −negative cases in early and advanced adenocarcinomas. The *p*-values were calculated by Wilcoxon rank sum test (two-sided, no multiple comparison adjustments) and are shown on top of each graph. n.s. not significant. Source data are provided as a Source Data file for (**a** and **b**).

repair pathways[31–33], whereas driver mutations in *EGFR* hinder such repair events[34]. Thus, in Early-Ad cases, *EGFR* mutations may cause the accumulation of a larger number of somatic mutations due to such disturbances. When the tumor progresses to Advanced-Ad stages, aberrations in other tumor-suppressor genes may appear even in *EGFR*-mutation negative cases, concealing the precedent patterns due to the initial *EGFR* mutations.

## Mutational signatures

We conducted a mutational signature analysis[35] and detected eight signatures for single-base substitutions (SBS) (Fig. 4a). Regardless of the status of tumor progression, the most common signature was SBS5, which is associated with aging. APOBEC-associated signatures, SBS2/SBS13, and SBS18, which is a reactive oxygen species (ROS)-associated signature, were detected at higher frequencies in Advanced-Ad cases than in Early-Ad cases (Fig. 4b). In addition, we detected eight indels (ID) (Fig. 4c) and three doublet-base substitution signatures (Supplementary Fig. S5). Among these, several ID6 mutations were detected in two Advanced-Ad cases (Fig. 4c). The ID6 signature represents a homologous recombination deficiency caused by *BRCA1/2* mutations. Consistently, corresponding *BRCA2* mutations were identified in two cases (AD09-116 and S7) where the ID6 signature was the most relevant. Upon analyzing these two cases for SVs, we found that SVs of <50 kb in length were dominant, consistent with a previous report describing this as a feature of *BRCA2*-deficient tumors (Fig. 4d)[36]. This feature was not detected in other Advanced-Ad cases with a similar total level of SVs (AD09-115 and S16). Another signature, DBS6, which is considered a signature of "unknown etiology," appeared in five cases with high mutation rates. Although no possibly causative genomic mutations were identified[37,38], these cases should be further assessed in detail. Additionally, SBS4, ID3 and DBS2, which are tobacco-associated signatures, were associated with smoking history in 36 cases (47%) (only adenocarcinoma cases are shown in Fig. 4e).

Interestingly, APOBEC/ROS-associated signatures were commonly observed in the Noguchi type B tumor, probably due the low exposure of such tumors to environmental factors. At later stages, the tumor is more severely subjected to stress, as the cancer begins to interact with the stroma triggered by structural alveolar collapse, initiating the progression toward invasive adenocarcinoma (see section "Transcriptomic analysis").

## Epigenomic analysis

Next, we performed epigenome analyses on the same specimens. First, we analyzed patterns of genome-wide DNA methylation rates with direct methylation call obtained from PromethION (Supplementary Table S4)[39]. Based on the methylation data of each cytosine base, we calculated average DNA methylation rates in 50-kb windows for the entire human genome to profile genome-wide methylation status (see Methods for details).

Tumors seemed to have a distinct degree of DNA methylation at each stage. AIS, MIA/Lepidic-Ad, and Advanced-Ad cases had median methylation levels of 78.7%, 77.1%, and 73.2%, respectively (Fig. 5a). Hypomethylation frequently occurred in MIA cases (Fig. 5a, b), whereas genomes of AIS cases still retained a normal-level high-methylation status (Supplementary Fig. S6). For example, a 50-kb window of chromosome 7 in case TSU-39 (MIA) showed lower DNA methylation level than its paired normal specimens and other cases (Fig. 5c). We also found that a number of large insertions and copy number changes (especially copy number loss events) were moderately associated with the degree of hypomethylation (Fig. 5d). The time when genome-wide hypomethylation is initiated should be the same as when the genome instability levels might more increase in MIA/Lepidic-Ad or later stages.

To further scrutinize the association, we focused on the methylation of transposable elements, such as LINE-1, Alu, and SVA. In normal cells, the transposition of these elements is generally repressed by DNA methylation, as observed in our datasets. We found that when DNA methylation was disordered in cancer cells, some transposable elements did not undergo DNA methylation (Fig. 5e) and LINE-1 regions became hypomethylated in cancer genomes in some Lepidic-Ad and advanced cases. The hypomethylation of these elements may have provided the necessary background for the aforementioned occurrence of SVs at this stage. Notably, methylation levels of two lung squamous carcinoma cases (case S11 and case S14) were very low, which may demonstrate that this type of cancer generally harbors a large number of SVs.

We also calculated the DNA methylation levels in other possibly functional domains, such as at CpG islands, CpG shores, promoters, and potential enhancer regions (Fig. 5f). We found that a downstream promoter of the *APC* gene was methylated in several cases even at early stages (Supplementary Fig. S6e–h). However, the methylation levels of most of these regions, which should play pivotal roles in regulating target genes, were not heavily affected, suggesting that gene expression changes are derived from changes in individual genes rather than global epigenomic changes.

## Transcriptomic analysis

We first characterized the transcriptomic changes using bulk short read RNA-seq data (Supplementary Fig. S7 and Supplementary Table S5). Gene Ontology enrichment analysis that compared AIS of Noguchi type A and B showed that the genes in "organophosphate biosynthetic process" were activated in Noguchi type A cases, indicating that metabolic processes might change to facilitate cancer cell survival and growth (Fig. 6a, b, and Supplementary Data S1). Interestingly, in Noguchi type B cases, a remarkable upregulation of immune and angiogenesis-related genes was detected (Fig. 6b). Deconvolution analysis using lung cancer single-cell RNA-seq (scRNA-seq) data[40] also indicated that the proportion of cytotoxic lymphocytes, including CD8 + T, NK and NKT cells, increased in the Noguchi type B stage (Supplementary Figs. S7d and S7e; the association of immune features with other omics factors[41] has been discussed in Supplementary Fig. S7). When Noguchi type B tumors typically begin to harbor a lepidic growth region having some fibrotic foci with alveolar collapse, the cancers begin to interact with their microenvironments, invoking immune responses. The observed gene expression changes may be essential during the occurrence of the most remarkable transition in terms of histological changes from in situ to invasive. At later stages when it progresses to MIA/Lepidic-Ad, the induced expression of genes related to more malignant states, such as those involved in extracellular matrix disassembly and microtubule-based movement, was more relevant (Fig. 6c, d and Supplementary Data S1), which may finalize the transcriptomic transition to advanced cancer.

To directly validate these transcriptomic events further, we first performed spatial transcriptome sequencing analysis using Visium (Fig. 6e, Supplementary Fig. S8, Supplementary Table S6, and Supplementary Data S2). We examined local expression patterns in three representative cases of Noguchi types A and B, and MIS/Lepidic-Ad cases. To consider overall transcriptome features as a "transcriptome signature" of the respective stages, we considered the collective gene expression of key genes (Supplementary Data S1). The expression levels of the signature genes for the above transcriptome features were, in fact, locally enriched in the regions of corresponding pathological features (Fig. 6f). These histological and molecular features were consistent with the gene expression patterns and further validated by single-cell resolution multiplexed fluorescence immunostaining the using PhenoCycler (Fig. 6g, Supplementary Fig. S9, and Supplementary Table S7). In TSU-20 (AIS, Noguchi type A), tumor cells showed a pure-lepidic growth pattern, but no severe hyperplasia of the alveolar septum was observed. In this case, high levels of Noguchi type A transcriptome signature spots were distributed in the tumor cell

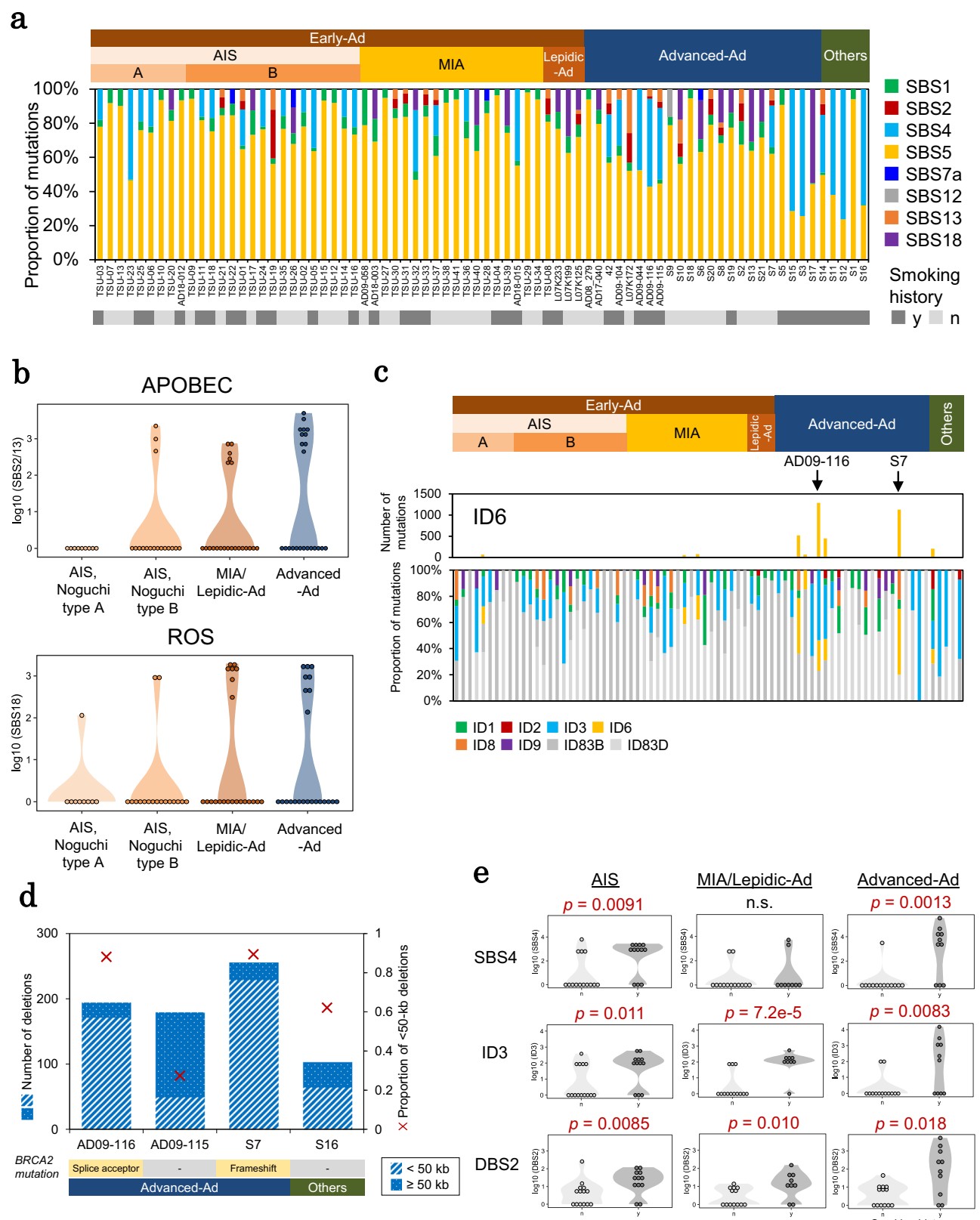

region (left panel, Fig. 6f). Few alveolar macrophages (AMs) and lymphocytes, if any, were infiltrated (Supplementary Fig. S9a) at locations with low Noguchi type A transcriptome signature scores. In TSU-21 (AIS, Noguchi type B), an example of the key stage of transcriptome changes, AMs and lymphocytes had migrated in the entire tissue region (regions of interest (ROIs)−1 and −2, Fig. 6g). This immune cell

infiltration was consistent with the Noguchi type B signature (middle panel, Fig. 6f). Notably, alveolar collapse with fibrotic foci (ROI-3, Fig. 6g) was observed, which is a histological characteristic of Noguchi type B tumors. We found that the *FOS/JUN*-highly-expressed cluster overlapped with the region of alveolar collapse (Fig. 6h). In TSU-33 (MIA, Noguchi type C), a region of partial fibrosis and invasion was

**Fig. 4 | Mutational patterns reveal mutagenic factors for cancer genomes.** **a** Proportion of single nucleotide substitutions (SBS) assigned to the COSMIC mutational signatures (v3.2). **b** Number of somatic mutations assigned to the APOBEC-associated signatures (SBS2 and SBS13) and the reactive oxygen species (ROS)-associated signature (SBS18) for each subtype in the upper and lower panels, respectively. **c** Proportion of indels (ID) assigned to the COSMIC mutational signatures (v3.2). Upper part of the graph, number of somatic mutations assigned to the ID6. Arrows indicate two cases with a high abundance of the ID6 signature. **d** Association between SV occurrence and *BRCA2* mutation status. The number of deletions and the proportion of shorter (<50 kb) deletions are represented in the graph. **e** Tobacco-related mutational signatures and smoking history of each case. Comparison of the number of somatic mutations assigned to tobacco-related signatures (SBS4, ID3, and DBS2) between smokers and non-smokers in each adenocarcinoma subtype. The *p*-values were calculated by Wilcoxon rank sum test (two-sided, no multiple comparison adjustments) and are shown in the top of each graph. n.s. not significant. Source data are provided as a Source Data file for (**a**, **b**, **c**, **d** and **e**).

observed (Supplementary Fig. S9b). This region is consistent with the region having a high score of the MIA/Lepidic-Ad signature (right panel, Fig. 6f).

To evaluate and validate the immune features of Noguchi type B tumors, we performed computational deconvolution analysis of Visium data of case TSU-21 using published scRNA-seq data[40] to characterize local immune cell profiles (Supplementary Fig. S8d). Interestingly, we found several types of macrophages, such as FABP4+ and SPP1+ macrophages (a subpopulation of CCL3L1+ macrophages[40]). Macrophages including SPP1+ macrophages were mainly located in the upper region (macrophage-rich region, Supplementary Fig. S8a). Macrophages highly expressing *FABP4* and *SPP1* were also densely present near the region of alveolar collapse (Supplementary Figs. S8a and S8e). The ligand–receptor interaction analysis revealed that SPP1 may be secreted from macrophages and received by integrin complexes in fibroblasts (Supplementary Fig. S8h), which indicted that SPP1+ AMs would affect fibroblasts in the vicinity and may cause alveolar collapse and fibrotic foci.

For further validation analyses at higher spatial resolution, we performed in situ gene expression profiling of 302 genes in TSU-20 (Noguchi type A) and TSU-21 (Noguchi type B) using Xenium (Supplementary Figs. S10 and S12, Supplementary Tables S8 and Supplementary Data S3). We obtained the expression levels of 58,648 and 351,742 cells in the entire tissue of these cases, respectively (Fig. 6i and Supplementary Fig. S12). Xenium data confirmed features of each cell, especially of AMs, at a single-cell level (Fig. 6j). Accordingly, we confirmed that FABP4+ and SPP1+ macrophages were localized in certain alveoli structures in the Noguchi type B case (Supplementary Fig. S11). These patterns were different from Noguchi type A where almost no SPP1+ macrophages were resided (Supplementary Fig. S12).

## Haplotype-aware analyses

We further performed genome analysis in a haplotype-aware manner. Taking advantage of the long read WGS data, we first "phased" the somatic mutations to their background germline heterozygous SNPs (see the Methods sections for more details and our previous study for technical evaluations[14]). In brief, we constructed phased blocks by performing haplotype phasing analysis of long read WGS data, and the somatic mutations detected in the cancer genomes were mapped to these phased blocks. An average of 5051 phased blocks were generated from the tumor genomes of Early-Ad cases, with an N50 block length of 1.2 Mb (Supplementary Table S9). An average of 31% (54% for variant allele frequency (VAF) ≥ 0.2) of point mutations and 26% of SVs were assigned to each haplotype, whereas the remaining 69% and 74% were assigned to haplotype-unknown regions or they remained undetected on long reads due to their low VAFs (Fig. 7a, b and Supplementary Fig. S13a).

With the obtained haplotype-aware data, we examined whether there are regions with enriched somatic mutations in either of the haplotypes. We found that point mutations were significantly enriched and biasedly distributed in both haplotypes (Fig. 7c). For example, of the ten SNVs significantly enriched in a 100-kb window (chr1: 95700001-95800000; adjusted *p* = 3.5e-8) in an AIS Noguchi type B case (TSU-01), nine were detected on haplotype #1 (HP1) (Fig. 7d). Interestingly, most SNVs were C > G substitutions, which is similar to

the APOBEC mutational signature (cosine similarity = 0.732). This result is similar to those obtained in our previous study of advanced cancers[14], where we discussed that a large chromosomal rearrangement event should account for the characteristic mutational signature. In this study, we detected that this type of event appeared first in the Noguchi type B stage of AIS.

We also found that the collected information could be utilized to elucidate the order in which somatic mutations occur, considering the relative frequencies of reads having either or both mutations. We found 94 mutation pairs for which the order of occurrence was resolved (Supplementary Fig. S13b). For example, in TSU-11, the order of occurrence of two SNVs (at a 4.8-kb distance apart) located in the LINC02147 region on chr5 was resolved (Fig. 7e). This result suggests that the A > C mutation occurred first, followed by the T > G mutation on the haplotype #2 in this region. This approach was particularly useful in Early-Ad cases. However, increased long read sequencing depth should enable further precise reconstruction of somatic mutation histories, especially for later cancer stages, where the accumulation order of each mutation has been completely recorded.

## Diverse clonal architectures of lung adenocarcinoma: from in situ to invasive phenotype

Lastly, we examined potentially diverse clonal structure of cancer cells, especially at early stages. We inferred the possible clone structure using PyClone-VI[42], based on the VAFs of the detected somatic mutations. As a result, multiple subclones were detected in 59 of 76 cases (78%) (Fig. 8a). The inferred clonal structure showed that the driver mutations, such as *EGFR* mutations, occurred in clones with the maximum cellular prevalence in almost all cases. These clones represent an early founder mutation in cancer cells, which would survive selective pressure until tumor cells eventually develop into invasive carcinomas.

Surprisingly, even at the AIS stage, ≥2 additional subclones were detected within the driver-mutation-harboring clone population for each case. Overall, the co-occurrence analysis of multiple SNVs performed using long read sequencing data supported the presence of multiple clones for at least 92 SNV pairs (Fig. 8b, 28 pairs were present in both short and long read WGS data).

Two subclones were detected in case TSU-02 (AIS, Noguchi type B) (Fig. 8c). The major clone harbored an *EGFR* exon 20 insertion as driver mutation. The minor subclones additionally harbored a *FOXA2* mutation. *FOXA2* is essential in lung epithelial cell differentiation and alveolar epithelium maintenance[43,44]. The observed *FOXA2* deficiency may have triggered the malignant transition[45–47] of the parental clone, which may be an essential factor in the progression from AIS to Lepidic-Ad. Consistently, the clone with a *FOXA2* mutation became a major one in Lepidic-Ad case L07K233. In TSU-01 (AIS, Noguchi type B), which had a driver mutation of *EGFR* exon 19 deletion and haplotype-biased APOBEC signature mutations which were reported above (Fig. 7d), both the *EGFR* mutation and the APOBEC signature were present in the dominant clone (Fig. 8d). In this case, even though the presence of multiple clones was detected by the PyClone analysis, two independent models for its clonal evolution were suggested: (1) the two clones came arose from a major clone in

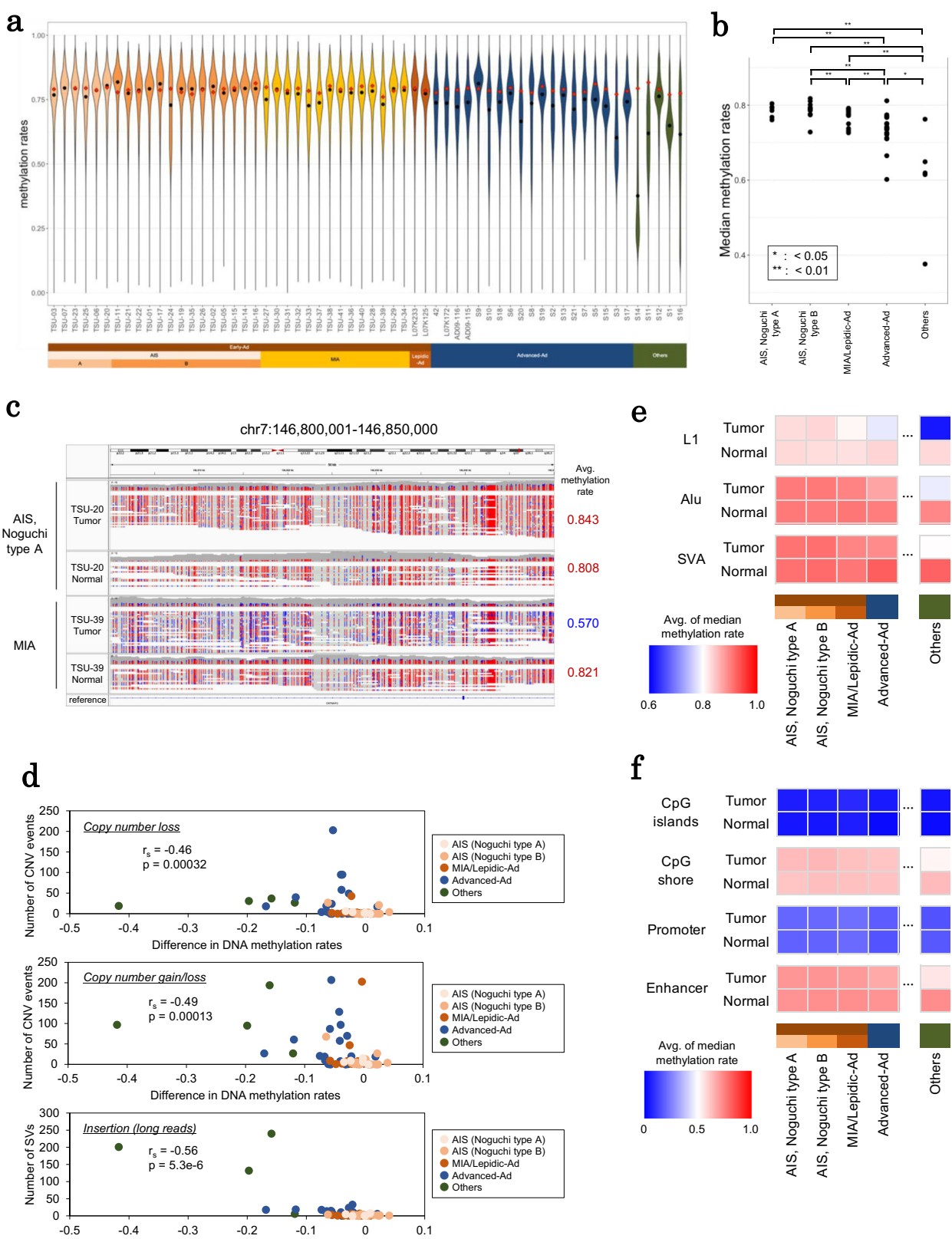

parallel, and (2) the third minor clone was generated from the second clone. Short read WGS data could not discriminate the order of somatic mutation occurrence. In addition, the lack of sufficient sequencing depth hampered the precise analysis. Therefore, increased sequencing depth for the long read analysis is needed for a more comprehensive detection of minor clones.

As the samples were independent of each other, we could not directly trace the evolutionary process from very early to advanced stages for each case. We could only infer the general history of cancer evolution by PyClone analysis and mutation phasing analysis. Nevertheless, we consider that the real evolutionary aspects of cancers should be represented by the sum of individual cancers, at least, to some extent.

**Fig. 5 | Epigenomic aberrations in early and advanced lung adenocarcinoma. a** Genome-wide methylation status of each case. The distribution of DNA methylation rates in each 50 kb genomic window is shown in the violin plot. DNA methylation rates of each normal counterpart shown as red dot plots. **b** Median genome-wide DNA methylation rates. The *p*-values were calculated by Wilcoxon rank sum test (two-sided, no multiple comparison adjustments). **c** An example of DNA hypomethylation. Unmethylated and methylated cytosine bases in CpG sites are colored in blue and red, respectively. **d** Association between differences in median genome-wide methylation rates (50 kb windows) from normal counterparts and number of copy number events and insertions. Each plot represents one case. Spearman correlation coefficient and *p*-values (two-sided, no multiple comparison adjustments) are shown in the inset. **e** DNA methylation patterns of LINE-1, Alu, and SVA in each subtype. **f** DNA methylation patterns of CpG islands, CpG shores, promoters, and possible enhancers in each subtype. Source data are provided as a Source Data file for (**b**, **d**, **e** and **f**).

## Discussion

In this study, we present a genome-wide and multi-modal long and short read sequencing characterization of lung adenocarcinoma from AIS to advanced cases (Fig. 8e). It is true that the number of samples collected for this study was not large. This was for the following clinical reasons. First, the number of patients who underwent surgery was relatively low: i.e., patients with advanced cancers were more likely to undergo surgery. Second, for the long read sequencing, it was necessary to extract DNA from fresh frozen AIS and MIA tissue specimens less than 2 cm in diameter and these were lepidic-type adenocarcinomas histologically, containing a smaller number of tumor cells, than advanced adenocarcinomas, most of which were papillary, acinar or solid-type adenocarcinomas. Therefore, it was not always easy to extract sufficient amounts of DNA from AIS and MIA specimens. However, even with this limited number of samples, we were able to characterize important features of early cancers as described here. Note that the described differences are supported by statistical tests (Figs. 1c and 5d).

We demonstrated that driver mutations already occur in AIS, whereas other cancerous aberrations mostly appear at later stages. We also found significant changes in environmental factors in AIS Noguchi type B, followed by dramatic changes in genomic aberrations in the invasive stage of lung adenocarcinoma. Thus, this stage is critical for developing lung adenocarcinomas. In this study, we focused on the molecular characterization of lung adenocarcinomas which were pathologically classified into AIS and MIA according to the Noguchi classification. Interestingly, there were different molecular features between Noguchi type A and B in AIS at the genome and transcriptome levels. Further, we found that at the Noguchi type B stage, characteristic changes make the cells more robust, allowing them to survive outside the alveoli. Further in-depth analysis of early-stage lung adenocarcinoma cells or other cancers should bring new insights into the initial mechanisms of cancer progression.

## Methods

### Clinical samples
Frozen surgical specimens from lung cancer patients were used in this study. Both tumor tissue and normal counterpart (peripheral lung) tissue were obtained separately from the same resected specimen. Normal counterpart tissue was carefully collected from the regions >5 cm from tumors with macroscopically normal morphology. The pathologists (HK and MN) confirmed using light microscopy that the normal counterpart tissue did not contain tumor cells. The smoking history of the cases was ascertained by interview using a questionnaire. A smoker was considered as a person who smoked regularly for ≥12 months at any time in their life, and non-smokers were those who did not. According to the WHO classification, lung tumors were diagnosed by cytological and/or histological examination. The institutional ethics committees approved in this study at the National Cancer Center (NCC), University of Tsukuba Hospital and the University of Tokyo (UT), Japan. All samples were obtained with the appropriate informed consent at the NCC and University of Tsukuba Hospital, Japan. The patient consent was obtained in a written form.

### Short read WGS
Short read WGS was conducted at both institutions. At the NCC, genomic DNA was extracted from frozen tumor tissues and paired normal counterparts using AllPrep DNA/RNA Mini Kit (Qiagen). According to manufacturer's protocols, library preparation was conducted using TruSeq DNA PCR-Free Library Prep Kit (Illumina). At UT, genomic DNA was extracted using MagAttract HMW DNA Kit (Qiagen). According to the manufacturer's protocols, library preparation was conducted using a TruSeq Nano DNA Library Prep kit (Illumina). Sequencing was performed using NovaSeq 6000 (Illumina).

### Long read WGS
High molecular weight genomic DNA was extracted from frozen tumor tissues and their normal counterparts using the MagAttract HMW DNA Kit (Qiagen). Library preparation was conducted using a Ligation Sequencing Kit (SQK-LSK109 or SQK-LSK112/114, ONT) according to the manufacturer's protocol. Sequencing was performed by PromethION (ONT) with R9.4.1 (FLO-PRO002) or R10.2/10.4 (FLO-PRO112/FLO-PRO114) flow cells.

### RNA-seq
Total RNA was extracted from frozen tissues using an AllPrep DNA/RNA Mini Kit (Qiagen). RNA-seq library preparation was conducted using a TruSeq Stranded mRNA Library Prep (Illumina), TruSeq Stranded Total RNA Library Prep Gold (Illumina), or a SMART-Seq Stranded Kit (Clontech Laboratories) (Supplementary Table S5). Sequencing was performed using a HiSeq 2000 system (Illumina).

### Spatial transcriptome sequencing
Fresh frozen tissues were sectioned at 10-μm thickness using a cryostat (CM1950, Leica). Methanol fixation and hematoxylin and eosin (H&E) staining were conducted according to a demonstrated protocol (CG000160, RevC, 10x Genomics). H&E imaging was performed using BZ-X800 (Keyence). Library construction was performed using Visium Spatial Gene Expression Slide & Reagent Kit (10x Genomics) according to the manufacturer's protocol (CG000239, RevF, 10x Genomics). Permeabilization time was set to 6 min. Sequencing was conducted by a NovaSeq 6000 system (Illumina).

### Multiplexed fluorescence immunostaining
Fresh frozen tissue sections were cut at 10-μm thickness using a cryostat (CM1950, Leica) and prepared in Poly-L-Lysine-coated coverslips (CS04863, Matsunami) or MAS-coated coverslips (CS04883, Matsunami). Multiplexed fluorescence immunostaining was conducted using the PhenoCycler system (Akoya Biosciences) according to the manufacturer's protocol (CODEX User Manual – Rev C) as follows. Briefly, the samples were dried by desiccant beads for 2 min and incubated with acetone for 10 min. At the humidity chamber, the samples were incubated for 2 min and tissue hydration was conducted with Hydration Buffer. The samples were fixed using Pre-staining Fixing Solution at a final concentration of 1.6% paraformaldehyde for 10 min. After washing the tissues using Hydration Buffer, the samples were incubated with Staining Buffer for 20 min. In the humidity chamber, the tissues were incubated with Antibody cocktail for 3 h. The information on antibodies is provided in Supplementary Table S7. After tissue staining, the tissue was washed by Staining Buffer twice and fixed by Post-Staining Fixing Solution for 10 min. The tissues were washed by 1× phosphate buffered saline (PBS) in three times and

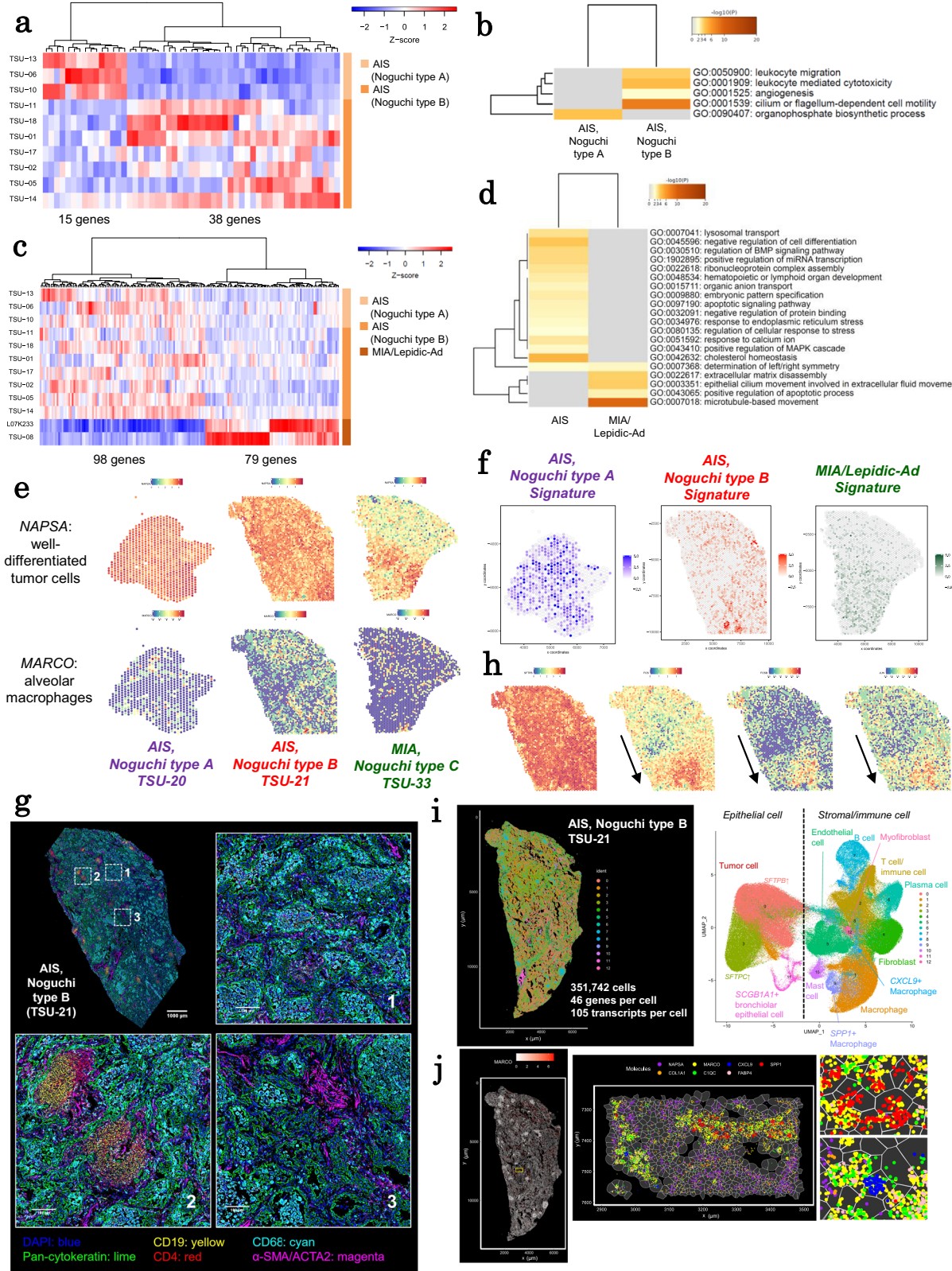

incubated in ice-cold methanol for 5 min. The tissues were also washed by 1× PBS in three times and fixed by Final Fixative Solution for 20 min in a humidity chamber. The samples were again washed by 1× PBS in three times and finally stored in Storage Buffer before running the assays. The reporter plate was prepared as shown in Supplementary Table S7. To run a multicycle fluorescence detection, a Keyence microscopy system (BZ-X800, Keyence) was used under the control by

CODEX Instrument Manager (CIM, Akoya Biosciences). The output image data was visualized by QuPath (version 0.3.2)[48].

**In situ gene expression profiling**

In situ RNA expression analysis at a single-cell level was performed using Xenium (10x Genomics). Fresh frozen tissue sections (10 μm thickness) were prepared on Xenium slides (10x Genomics) using a

**Fig. 6 | Transcriptome patterns in early and advanced lung adenocarcinoma. a** Expression patterns of DEGs between AIS Noguchi type A and B in RNA-seq data. **b** GO terms enriched in the DEGs in (**a**). **c** Expression patterns of DEGs between AIS and MIA/Lepidic-Ad cases in RNA-seq data. **d** GO terms enriched in the DEGs in (**c**). **e** Spatial transcriptome analysis Visium for three representative cases (TSU-20, TSU-21, and TSU-33). Expression patterns of marker genes for well-differentiated tumor cells (*NAPSA*) and AMs (*MARCO*) are represented. **f** Enrichment scores of transcriptome signature genes in the corresponding histological types in Visium data. **g** Multiplexed fluorescence immunostaining (PhenoCycler) of serial sections of spatial transcriptome data for case TSU-21. Protein expression patterns of representative cell-type markers for the entire region and three regions-of-interests: (1) a macrophage-rich region, (2) a region with lymphocyte infiltration, and (3)

a region nearby fibrotic foci characteristic to Noguchi type B tumors. Each antibody was used as a cell-type marker: Pan-cytokeratin, epithelial cells; CD19, B cells; CD4, T cells; CD68, macrophages; and α-SMA/ACTA2, myofibroblasts. **h** Spatial expression patterns of tumor cell markers in Visium data. **i** In situ gene expression profiling Xenium for case TSU-21. Clusters are represented in the spatial plot and the UMAP plot in the left and right panels, respectively. The statistics of Xenium data are shown in the margin of the spatial plot. The cell-type annotation is shown in the margin of the UMAP plot. **j** Xenium spatial expression pattern of an alveolar macrophage marker *MARCO* (left). Patterns of several macrophage markers in local regions (middle and right). Each dot represents a detected RNA molecule.

cryostat (CM1950, Leica). Fixation and tissue permeabilization were performed according to the protocol (CG000581, Rev A, 10x Genomics). Probe Hybridization Mix was prepared using a pre-designed panel (Xenium Human Lung Gene Expression panel, developmental build) and custom panel (Xenium Custom Gene Expression panel, design ID: 9GT3BT) according to the user guide (CG000582, Rev A, 10x Genomics). The list of 302 target genes is provided in Supplementary Data S3. The prepared probes were hybridized at 50 °C overnight. Post-hybridization wash (37 °C for 30 min), ligation (37 °C for 2 h), and amplification (30 °C for 2 h) were performed according to the user guide. Autofluorescence quenching and nuclei staining were conducted in the dark. Using the prepared slide, fluorescent probe hybridization and imaging were conducted using the Xenium Analyzer (on-board analysis: version 1.1.0.2, software: version 1.1.2.4, 10x Genomics). The output images and expression profiles were evaluated by Xenium Explorer (version 1.1.0, 10x Genomics). After the Xenium run, H&E staining was performed on the Xenium slide. For quality control of Xenium analysis, we checked the output summary HTML file and confirmed that the number of detected transcripts is compatible with that in the literature[49] reported by 10x Genomics.

**Target captured sequencing**

Using the extracted genomic DNA samples, target enrichment and library preparation were performed using SureSelect XT HS Reagents (cat# G9702C, Agilent Technologies) and SureSelect XT NCC oncopanel (cat# 931197, Agilent Technologies). The generated libraries were sequenced by NovaSeq 6000 system at 150 bp paired-end reads.

**Detection of point mutations**

Whole-genome sequences were mapped to the human reference genome hg38 using BWA-MEM (version 0.7.17)[50]. The mapped reads were sorted and indexed by SAMtools (version 1.9)[51,52], and duplicate reads were marked by Picard MarkDuplicates (version 2.20.7). Somatic mutations were detected using GATK Mutect2 and FilterMutectCalls (version 4.1.3.0)[53]. The somatic mutation VCF files were annotated by using ANNOVAR[54].

For Fig. 1b, cancer-related genes were defined as oncogenes: *EGFR*, *KRAS*, *NRAS*, *HRAS*, *BRAF*, *ERBB2*, *MET*, *MAP2K1*, *PIK3CA*, *RET*, *ROS1*, and *ALK*; tumor-suppressor genes: *TP53*, *PTEN*, *NF1*, *CDKN2A*, *CDKN1A*, *BRCA1*, *BRCA2*, *RB1*, *ATM*, *STK11*, and *APC*, and others: *ARID1A*, *ARID1B*, *ARID2*, *SMARCA4*, *SMARCA2*, *U2AF1*, and *RBM10*[20,55]. No known driver, nonsynonymous, or splice site mutations were detected in *HRAS*, *MAP2K1*, *ROS1*, *NF1*, *CDKN1A*, *RB1*, *ATM*, and *U2AF1* are not shown in the figure. In three cases (TSU-02, TSU-07, and TSU-26), *EGFR* hotspot mutations, which were filtered out by FilterMutectCalls, were manually picked.

To confirm the presence of driver mutations, target captured sequencing data with sufficient depth for 36 cases were used (Supplementary Table S10). Two driver mutations (*KRAS* G12D in AD09-058 and *EML4-ALK* in L07K125) were only detected from the target captured sequencing data because of the low tumor purity of these cases.

**CNV detection**

CNVs were detected from WGS datasets of tumor and normal pairs using Control-FREEC (version 11.6)[56,57] with the parameters "ploidy = 2," "window = 5000," "minimalSubclonePresence = 20." The mappability information (out100m2_hg38.gem) was included as input. Copy number gain and copy number loss events were defined as ≥4 and ≤1 copy copies, respectively.

For Fig. 1d, copy number gains in the following genes are shown: oncogenes including *EGFR*, *KRAS*, *ERBB2* and *MET* and in genes whose amplifications were frequently observed: *MYC*, *MYCN*, *MYCL*, *NKX2-1*, *TERT*, *MDM2*, *CCNE1*, *CCND1*, *CDK4*, and *CDK6*[20,24]. Copy number loss events in the following genes are shown: tumor suppressor genes: *TP53*, *PTEN*, *NF1*, *CDKN2A*, *CDKN1A*, *BRCA1*, *BRCA2*, *RB1*, *ATM*, *STK11*, and *APC*[20,55]. No copy number gains were detected in *MYCN* and no copy number losses were observed in *CDKN1A*, and are thus not shown in the figure.

**SV detection**

From long read WGS data, whole-genome long read sequences were mapped to the reference genome hg38 using Minimap2 (version 2.17-r941)[58] with the options "-ax map-ont" and "--MD." After removing secondary alignments, Nanomonsv (v0.5.0)[27] was used to detect somatic SVs and classify long insertions. To detect SVs, the options "--use_r-acon," "--var_read_min_mapq 20," "--min_tumor_variant_read_num 3," and "--min_tumor_VAF 0.01" and a normal panel constructed using all other normal WGS data was used when running "nanomonsv get." "nanomonsv insert_classify" was used to annotate inserted sequences.

From the short read WGS data, somatic SVs were detected by GenomonSV (version 2.6.1) using the reference genome hg19. GenomonSV filt was performed with the options "--min_junc_num 1" and "--matched_control_bam."

To validate the SVs detected in short read WGS data, we evaluated whether the long reads covered SV junctions using NCBI BLAST+ (version 2.13) as follows: (1) SV junction sequences were created with 500 bp adjacent sequences from each junction and the inserted sequence. The database was created by makeblastdb. (2) Long read sequences (query) were aligned to the constructed SV junction sequences (subject) using blastn with the option "-perc_identity 80." The SV junction sequences covered by long reads (≥90% coverage and ≥90% identity) were extracted as validated SVs.

For Fig. 1f, to detect whole-gene deletion of the *CDKN2A* region, SV breakpoints of deletions which completely covered the *CDKN2A* genic region were extracted.

**Mutational signature analysis**

From the VCF file of somatic point mutations, mutational signatures were extracted and decomposed into known COSMIC mutational signatures (v3.1)[35] using SigProfilerExtractor (version 1.1.4)[59].

**DNA methylation analysis**

DNA methylation status was profiled and calculated from the FAST5 files of long read WGS data from PromethION. For the data from R9.4.1 (FLO-

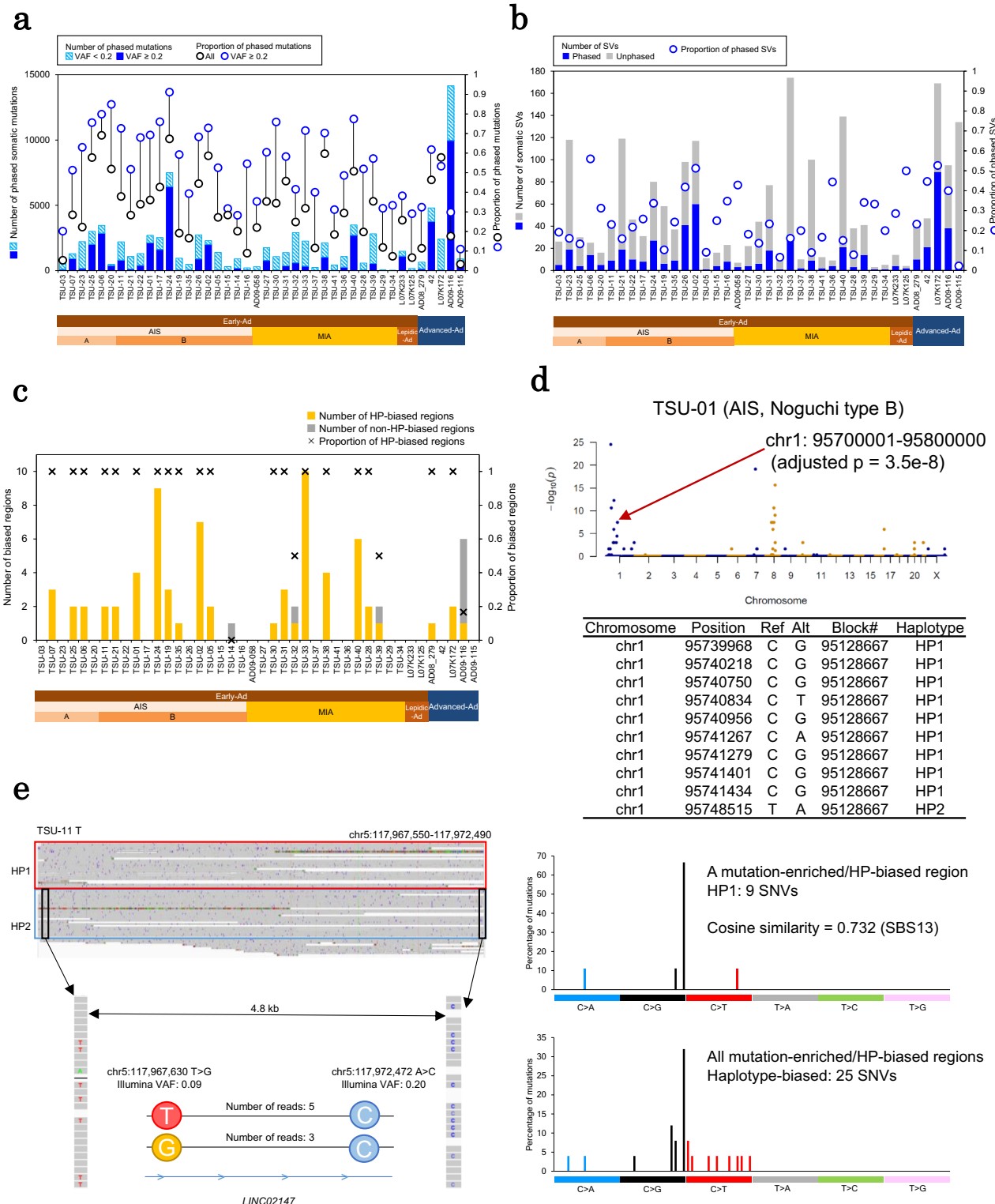

**Fig. 7 | Haplotype-resolved genomic, epigenomic, and transcriptomic aberrations in early and advanced lung adenocarcinoma. a** Number and proportion of somatic point mutations assigned to haplotypes. **b** The number and proportion of SVs assigned to haplotypes. **c** Number and proportion of mutation-enriched and haplotype-biased regions. HP: haplotype. **d** An example of mutation-enriched and haplotype-biased regions in an AIS case. The graph shows all mutation-enriched windows in this case (top panel). The *p*-values were calculated by hypergeometric

test using R function phyper (one-sided, adjusted by Bonferroni correction). List of somatic mutations in a mutation-enriched and haplotype-biased region (middle panel). The 96 substitution patterns of somatic mutations in the mutation-enriched and haplotype-biased regions (bottom panel). **e** An example of the somatic mutation pairs for which long reads directly resolve the occurrence order. Source data are provided as a Source Data file for (**a**, **b**, **c** and **d**).

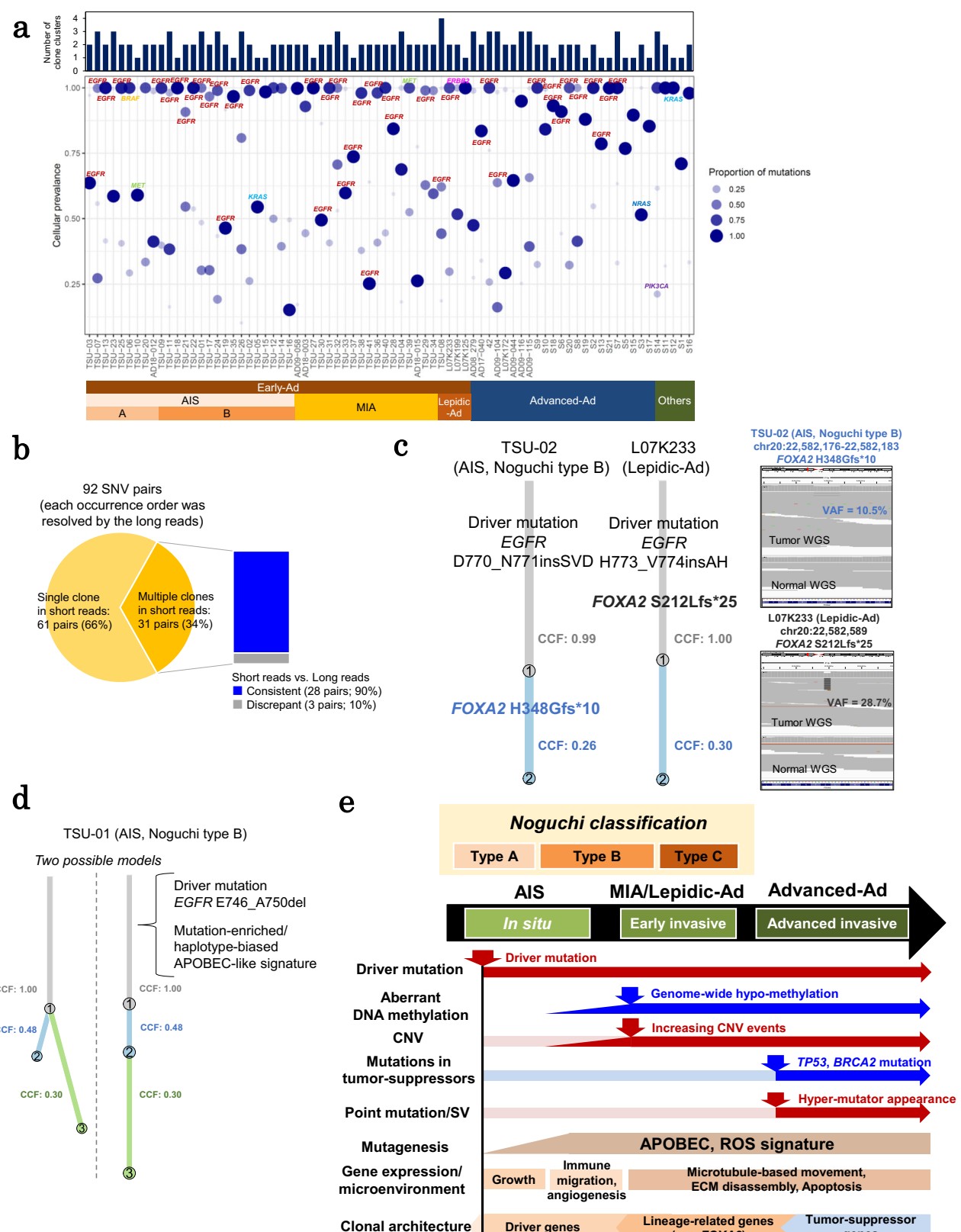

**Fig. 8 | Clonal architecture of early adenocarcinoma. a** Mutation clusters (clones) in all cases. The dots represent each clone and their size indicates the proportion of somatic mutations in the clone. The number of clones is shown in the bar graph. **b** Validation of estimated clone structures using long reads. The occurrence order of a total of 92 SNV pairs was directly resolved by long reads. The 61 SNV pairs were assigned to the same single clone population of the PyClone result. The breakdown (consistent/discrepant with the evaluation by the long reads) of the 31 SNV pairs assigned to the different multiple clones of the PyClone result were represented in the bar chart. **c** Estimated clone structures in two cases, TSU-02 (AIS Noguchi type B) and L07K233 (Lepidic-Ad), which harbored *FOXA2* mutations. CCF cancer cell fraction. **d** Two candidates of clone structures in case TSU-01 (AIS Noguchi type B). **e** Summary of genome-wide and multi-modal characterization of lung adenocarcinomas from early to advanced cases. Source data are provided as a Source Data file for (**a**).

PRO002) flow cells, methylation status at CpG sites was calculated using Nanopolish (version 0.13.2)[39,60]. The BAM files generated by Minimap2 were converted by nanopore-methylation-utilities[61] for visualization by IGV[62] in the bisulfite mode. For the data from R10.4 (FLO-PRO112/FLO-PRO114) flow cells, we used guppy (version 6.2.1, ONT) for DNA methylation calling with dna_r10.4_e8.1_modbases_5mc_cg_sup.cfg or dna_r10.4_e8.2_modbases_5mc_cg_sup.cfg models.

## Gene expression analysis using RNA-seq data

Datasets prepared using TruSeq Stranded mRNA Library Prep Kit were only used to perform expression analysis (Poly-A Method, Supplementary Table S5). Paired-end reads were mapped to the human reference genome hg38 using STAR (version 2.7.5c)[63] after trimming adapters using fastp (version 0.23.2)[64] and removing ribosomal RNA sequences using Bowtie 2 (version 2.3.4.3)[65]. The raw count and RPKM values were calculated using featureCounts (version 2.0.2)[66]. Differentially expressed genes (DEGs) among histological classes were extracted using DESeq2 (version 1.32.0)[67] with the Wald test (adjusted $p < 0.1$). Gene enrichment analysis was performed using Metascape[68].

## Analysis of spatial transcriptome data

The obtained Visium sequencing data and the H&E images were processed by Space Ranger (version 1.3.1, 10x Genomics). Data visualization and further analyses were performed using Seurat (version v4.3.0)[69]. Normalization was performed using the SCTransform function. Clustering analysis and UMAP were performed using the first 30 principal components (PCs). DEGs were extracted in each cluster and used for cluster annotation. Enrichment analysis of the signature genes (Supplementary Data S1) was performed by PAGE enrichment using Giotto (version 1.1.2)[70]. Histological annotation was conducted by pathologists using H&E images of fresh frozen sections using Visium and FFPE sections from the same cases. Ligand–receptor interaction analysis was conducted using CellChat (version 1.6.1).

## Analysis of in situ gene expression data

The output files of Xenium Analyzer were further processed using Seurat (version 4.3.0). The data was normalized using SCTransform. Clustering analysis was conducted using the first 30 PCs by FindNeighbors and FindClusters (resolution set to 0.3) functions. UMAP was also performed using the first 30 PCs. The top 10 genes which showed differentially higher expression levels in each cluster were identified using the FindAllMarkers (only.pos = TRUE, min.pct = 0.25, logfc.threshold = 0.25) function.

## Deconvolution analysis

For RNA-seq data, the reference expression pattern of each cell type was constructed using the scRNA-seq data of lung cancers[40]. Using Seurat, clustering was performed, and 26 cell-type clusters were extracted and annotated, as shown in Supplementary Fig. S7c. Deconvolution analysis was conducted using CIBERSORTx (version 1.0)[71].

For Visium data, deconvolution analysis was performed using two reference datasets (scRNA-seq[40] and Xenium) using RCTD[72] of R package spacexr (version 2.2.1). The results were visualized using R package scatterpie (version 0.1.9).

## Phasing analysis

Phasing analysis of tumor and normal genomes was conducted as reported in ref. 14. Briefly, germline SNPs were detected using GATK HaplotypeCaller (version 4.1.3.0) with base quality score and variant quality score recalibration from the short read WGS of normal samples. SNPs with PASS filter were extracted using BCFtools (version 1.9)[73]. Next, phasing analysis of the mapped long read WGS data of tumor and normal samples was conducted using WhatsHap phase (version 1.1)[74,75] with the obtained germline SNP information. For the long read BAM files, haplotype tags were added using WhatsHap haplotag.

## Haplotype assignments of somatic mutations

Somatic point mutations were assigned to each of the two haplotypes according to long reads with haplotype tags (the result of WhatsHap analysis) as follows[14]: (1) Long reads were identified with point mutations and extracted using SAMtools (version 1.12) mpileup. (2) Haplotype information was assigned to the mutation when ≥3 mutant long reads harbored one haplotype tag (HP1 or HP2) and ≤1 mutant read harbored the opposite haplotype tag (HP2 or HP1). We defined these point mutations as "phased" mutations.

The genomic regions where somatic mutations were significantly enriched and biasedly located in one haplotype were defined as follows[14]: (1) 100 kb windows were extracted from the whole genome region. (2) The statistical significance of somatic mutation enrichment in each region was evaluated using hyper-geometric distribution (R phyper function). Windows with a Bonferroni adjusted $p$-value < 0.1 were extracted as "mutation-enriched windows." (3) Overlapping regions between the mutation-enriched windows and the phased blocks were extracted. Overlapping regions where ≥50% of the somatic mutations were phased and ≥4 phased somatic mutations were located were considered for further analysis as "mutation-enriched regions." (4) Regions where ≥80% of the phased somatic mutations were biasedly assigned to one haplotype were defined as "mutation-enriched and haplotype-biased regions."

SVs were also assigned to haplotypes according to the supporting reads of the breakpoints[14]. For primary alignment, haplotype tags of the supporting reads were directly extracted from the haplotype-tagged BAM file using SAMtools mpileup. For supplementary alignment, the haplotype information was manually assigned by phased SNP information. Supplementary alignment reads were assigned to the haplotypes when ≥2 phased SNPs were detected and ≥70% of these SNPs were located on one haplotype. We assigned each SV to the haplotype when ≥3 supporting reads were assigned to the haplotype, and ≥70% of these supporting reads harbored one haplotype tag (HP1 or HP2). We defined these SVs as "phased" SVs.

## Estimation of clone architecture

Clone numbers were estimated according to VAFs of somatic point mutations using PyClone-VI (version 0.1.1)[42] with the parameters -c 40 -d beta-binomial -r 10. The input files included read count information of somatic mutations from the Mutect2 results and information on copy numbers and cellularity, which were calculated from tumor and normal WGS datasets using FACETS (version 0.6.2)[76]. From the BAM files, read count files for dbSNP build 151 (https://ftp.ncbi.nlm.nih.gov/snp/organisms/human_9606/VCF/common_all_20180418.vcf.gz) were generated by snp-pileup (version 434b5ce) with the parameters "-g -q15 -Q20 -P100 -r25,0" according to the developer's tutorial. Next, copy numbers were calculated using FACETS with "cval = 400" for procSample function. The clone evolution of each case was inferred and visualized by ClonEvol (version 0.99.11)[77].

## Extraction of the order of mutation occurrences

For pairs of two somatic mutations, their occurrence order was resolved using long reads which directly covered both positions of the two mutations[14]. If a mutation with a lower VAF was fully detected with the other mutation with a higher VAF in the same long reads (double-mutant reads), the mutation with a higher VAF occurred first, and these mutation pairs were assigned to the same haplotype.

To perform this analysis, we evaluated whether the detected somatic mutations from the short read WGS existed in haplotype-tagged long reads using SAMtools (version 1.12) mpileup. For determining the order in each mutation pair, ≥3 single-mutant reads where

only the mutation with a higher VAF was detected and ≥3 double-mutant reads without inconsistencies needed to be extracted.

## Statistics and reproducibility
Information on statistical testing was provided in the description of each test in *Methods* section and the corresponding figure legends. No statistical method was used to predetermine sample size.

## Reporting summary
Further information on research design is available in the Nature Portfolio Reporting Summary linked to this article.

## Data availability
The raw sequencing data including short read WGS, long read WGS, RNA-seq and spatial transcriptome data from the adenocarcinoma cases have been deposited in the Japanese Genotype-Phenotype Archive (JGA, http://trace.ddbj.nig.ac.jp/jga), which is hosted by the National Bioscience Database Center and DDBJ under accession code JGAS000570. These data are available under restricted access due to ethical restriction. The raw sequencing data of short read WGS and long read WGS from the 20 advanced NSCLC cases that were previously obtained and reported[13,14] were deposited in the DDBJ JGA with accession numbers JGAS000065 (JGAD000252 and JGAD000253) and JGAS000349, which are available under restricted access due to ethical restriction. These raw sequencing data are under restricted access because the Act on the Protection of Personal Information in Japan defines them as personally identifiable information. To access these data, users require the approval by the NBDC (https://humandbs.biosciencedbc.jp/en/guidelines/data-sharing-guidelines). The users can apply for the use of data via the application system (https://humandbs.biosciencedbc.jp/en/data-use). The restrictions for granting data are described in the following URL (https://humandbs.biosciencedbc.jp/en/guidelines/security-guidelines-for-users). The processed data was deposited in the database DBKERO (https://kero.hgc.jp/)[78] and made publicly available in the download page (https://kero.hgc.jp/Early_cancer.html). The human reference genome hg38 was downloaded from the UCSC Genome Browser (https://hgdownload.soe.ucsc.edu/downloads.html). Source data of the figures are provided with this paper. Source data are provided with this paper.

## Code availability
The code used in this study is available in the GitHub repository at https://github.com/asuzuki-asuzuki/Early-Ad_2023.

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

## Acknowledgements

We thank K. Abe, M. Tsubaki, K. Imamura, M. Satake, Y. Kanayama, Y. Kuze, E. Sekimori, E. Ishikawa, R. Fujinaga, S. Takashima, Y. Shimada, M. Matsuda, and H. Wakaguri for their technical assistance. We are grateful to National Cancer Center Biobank. This work was supported by JSPS KAKENHI Grant Number 22H04925 (PAGS) (to Y. Suzuki.) and 20H00545 (to T.K.). This work was supported by Japan Agency for Medical Research

and Development (AMED) P-PROMOTE grant number JP21cm0106582 (to A.S.), JP23ama221522 (to A.S.) and 21cm0106577 (to T.K.). The supercomputing resource was provided by Human Genome Center, the University of Tokyo (http://sc.hgc.jp/shirokane.html).

## Author contributions

M.N., T.K. and A.S. designed the study. K.S., Y.Suzuki. and T.K. conducted sequencing. Y.H., Y. Sakamoto., K.K., M.O., M.Seki., Y.Shiraishi. and A.S. contributed to the analysis of the sequencing data. M.A., J.Z. and A.S. conducted spatial transcriptome, multiplexed fluorescence immunostaining and in situ gene expression profiling analysis. H.K., N.M., M.Shirasawa., M.Y. S.I.W., D.M., M.N. and T.K. coordinated the samples. H.K., N.M., Y.Y., D.M. and M.N. performed histopathological evaluation. M.Seki., A.K., Y.Suzuki., M.N. and T.K. interpreted the findings and supervised the study. Y.H., Y.Sakamoto., K.K., Y.Suzuki., M.N., T.K. and A.S. wrote the manuscript. Co-authors include female and male researchers and Ph.D. candidates. All the authors are Japanese, but this is due to the condition of the informed consent of the patients, which assumes the domestic use of the material and data. All authors have approved the final version of the manuscript.

## Competing interests

The authors declare no competing interests.
