## [Peer Review File · Nature Communications]

REVIEWER COMMENTS

Reviewer #1 (Remarks to the Author): Expert in genomics, bioinformatics, and long-read sequencing

NOTE FROM THE EDITOR: This Reviewer provided all their comments in the attached file.

Reviewer #2 (Remarks to the Author): Expert in spatial transcriptomics, cancer transcriptomics, and tumour microenvironment

The manuscript describes the results of whole-genome sequencing analysis of 76 lung cancer specimens, including 48 early and small-sized lung adenocarcinoma cases and 8 cases of advanced adenocarcinomas. The sequencing data were analyzed to detect somatic mutations, including point mutations, SVs, and CNVs. The analysis revealed that EGFR driver mutations were acquired early in 63% of the Early-Ad cases, and the frequencies of EGFR mutations among the Early-Ad cases were similar between the different pathological classes and the advanced cases. The driver compositions observed were broadly similar to those observed in advanced adenocarcinomas, suggesting that the most important key driver mutations occur at early stages.

This manuscript offers an important addition of a novel cohort of early-stage lung cancer lesions to the existing large databases of lung adenocarcinoma of the TCGA and ICGC.

It could benefit from a more comprehensive discussion of clinical & diagnostic implications. I found the biological contextualization to be underwhelming.

The rich resource and the wealth of performed molecular assays is important for the community, but reuse of the dataset will require facilitated access to the repository, availability of processed datasets, and the upload of well-annotated analysis code.

Major comments:

Data availability: I was not able to assess the completeness of the generated datasets due to the controlled-access criteria of the used repository. The authors report in their "reviewer's token file" that all the raw data has been made publicly and freely available. I was not able to complete the lengthy

NBDC application within the scope of this review and would appreciate an actual reviewer's token for easier access to the raw data. Additionally, I expect that the authors also provide the final processed files to facilitate the reproduction of the reported results. Processed data (excluding raw sequencing reads) could be uploaded to an open-access repository.

Reproducibility of computational analyses: The paper relies on extensive computational analyses, yet the provided computational methodologies are superficial and do not enable independent reproduction of the analyses. For reproducibility purposes, the authors need to provide access to the underlying code in an appropriate repository (ideally with DOI, eg Github+Zenodo).

Spatial transcriptomic analyses: the loupe browser is suitable for exploratory visualizations but not for reproducible analyses and for the generation of publication-quality figures. The authors should use one of the many suitable coded analysis pipelines for visium analysis to ensure reproducibility.

The reporting of the transcriptomic analyses seems incomplete. The authors jump straight to a DGE of Noguchi type A vs type B and report the enriched GO terms. It would be of interest to see a PCA plot of all the generated transcriptomic datasets to better understand the landscape of the assessed samples.

Bulk RNA samples might be confounded by the presence or lack of surgical margins, infiltration of other cell types and handling time after surgery. The authors use Visium to validate bulk RNA sequencing. Neither of these methods enable single cell resolution. Some of the changes are then validated with imaging data (single cell resolution). Have the authors considered computational deconvolution or mapping to existing single cell RNA sequencing datasets for this disease entity?

Minor:

Increase quality/resolution of the plot and text in Figure 7e. Currently, the legend is not legible upon enlargement.

Increase quality/resolution of Visium plots in Supplementary Figure 6. Currently, the legend is not legible upon enlargement.

Figure 8c right plot: background plot seems to be a low-quality screenshot, please update

Reviewer #3 (Remarks to the Author): Expert in lung cancer genomics and evolution

Thanks for the opportunity reviewing this interesting study by Haga etc. on the molecular evolution during early carcinogenesis. This is the first study using long read WGS and spatial gene expression analyses on this topic that I am aware of. Therefore, the data is welcomed to the field. However, the study still has some important limitations as listed in the following.

1. The sample size is still pretty small. It is understandable that these lesions may not be as many as invasive diseases. The authors should discuss this important limitation and the results should be interpreted with caution.
2. The authors used multiomics approaches, which is an important strength of this study. However, it is not easy to follow which samples had short read, long read, Visium, bulk seq, mIF etc. It will be helpful to have a schema of the study to illustrate the information and the rationale for certain assays. For example, how many samples went for Visium? How did you decide to choose samples for Vixium etc.
3. Most findings have been reported in the previous studies, therefore, these results are confirmatory. I would suggest the authors put their cards down and emphasize the novel findings in this study in the abstract and in the summary/conclusions.
4. The association between EGFR mutations and TMB, CNV burden etc. is interesting. Did the authors control for smoking status? One plausible explanation the authors presented is the association of EGFR mutations with DNA repairs. Can authors analyze their own data, mutations or gene expression-based pathway analysis between EGFR WT vs EGFR mutant to provide the evidence to support this?
5. The authors have generated a lot of interesting data, which can be dug in deeper. For example, deconvolution can be applied to RNA seq data to look into the immune infiltration by TIMER, CIBERSORT, etc. How many cases underwent Visium profiling? These spatial gene expression data can be explored in a much deeper way such as difference between different histological subtypes; between tumor, tumor margins, distant normal lung; cell-cell interaction; ligand-receptor analyses etc.
6. The immune features during early lung cancer development are important. There are recent publications in this space. I would suggest the authors to dig more by including more mIF data or at the minimum deconvolute the RNA seq data to address this important aspect and discuss their findings in the setting of recent publications on this topic. There are two publications at least on Nature Communications by Chen et. al in 2019, PMID: 31784532 and the other one by Dejima et al PMID: 33976164.
7. The authors can do a better job integrate these multiomics data.

8. Finally, the term evolution in the title and body of the manuscript may not be appropriate. Although these may be different stages during lung cancer evolution, all these specimens are resected specimens. They are independent to each other and cannot be used to infer the evolutionary process as they are not related. All observations are association that may suggest the evolutionary relationship. This is a universal issue in the field of cancer evolution research. But the authors should discuss this limitation.

The authors performed multi-omics experiments, including short-read WGS, long-read WGS, short-read RNA-seq, and spatial transcriptome sequencing on 76 lung cancer samples (48 early-stage cases, 20 advanced-stage cases, and 8 others) and their matched normal samples. By performing analyses including profiling somatic SNV, CNV, SV, haplotype-specific SNV/SV, DNA methylation, mutation abundance, mutational signatures, gene expression, and clonal architecture reconstruction, the major conclusions reported by authors include 1) the driver mutations already occur in AIS, whereas other cancerous aberrations majorly appear at later stages; 2) significant changes in environmental factors impacted AIS Noguchi type B, followed by dramatic changes in genomic aberrations in the invasive stage of lung adenocarcinoma. 3) there were different molecular features between Noguchi type A and type B in AIS at the genome and transcriptome levels.

I have the following comments regarding the methods, results, and academic writing:

Major comments:

1. How the driver mutations are obtained is not clearly stated. Although EGFR mutation is apparently obtained from previous publications (10, 11, and 15), other driver mutations such as MET, KRAS, *etc* need sources (lines 174-176 on page 7). To make the claims solid, it is required to add references or other evidence to both the main text and Table S1, where driver mutations are listed.
2. The somatic mutations are apparently obtained by comparing a tumor sample and the matched “normal” sample from the same individual. However, details about how the “normal” samples are defined and obtained are not stated in the manuscript. This is critical for reproducibility. It is also recommended to clearly state that “mutation” in this work means “somatic mutation” to avoid misunderstanding for the general audience.
3. The claim that “AIS cases are still free from mutations in the tumor-suppressor genes, especially TP53” lacks statistical tests (lines 185-186 on page 7). One of the evidence supporting this conclusion is 7 out of 48 Early-Ad cases harbored relevant mutations in tumor-suppressor genes while 11 out of 28 (28 = 76 - 48) Advanced-Ad cases “harbored deleterious mutations in tumor-suppressor genes”. A statistical test is needed since the sample size is not very large. I did a quick Fisher Exact Test and the p-value is about 0.024, which is barely significant. The mutations on tumor-suppressor genes in Advanced-Ad are not “far more” frequent than that in Early-Ad cases as claimed by the authors if randomness is considered. The other evidence is Fig 1c, which provides gene-by-gene mutation frequencies. Adding a statistical test for each gene is required for a solid conclusion.

The R (version 4.2.2) code for Fisher Exact Test is:

```
dat <- matrix(c(7, 48 - 7, 11, 28 - 11), nrow = 2, dimnames = list(c("mutation present",  
"mutation absent"),c("Early-Ad", "Advanced-AD")))  
fisher.test(dat, conf.int = FALSE)
```

4. I am not convinced by the claim “Exceptionally, long-range deletion and copy number loss

events of CDKN2A/p16 region were identified even in Early-Ad cases (Fig. 1f and Supplementary Fig. S1), which is consistent with a previous report, collectively suggesting that the key driver gene mutations and this gene deletion should be essential to initiate the early cancer.” (lines 192-196 on page 7) for three reasons:

- 1) Logically, a mutation (SNV + CNV, etc) event present in the early stage does not prove that it is essential to initiate early cancer. Evidence that the absence of this mutation inhibits the initiation of early cancer is needed to make this claim.
 - 2) Fig 1f does not include any information about “long-range deletion and copy number loss events of CDKN2A/p16 region”.
 - 3) It is not clear that there is a deletion in the CDKN2A region based on the IGV snapshot of Supplementary Fig. S1a. In addition, the color code in the figure is not explained.
-
5. In line 218 of page 8, “... driver fusion KIF5B-RET....” while in Fig 2b it is “KIF5B-RET inversion”. The difference should be explained or corrected if it is a typo. In addition, the IDs/names of the truncated reads in KIF5B and RET region should be provided to prove there is a gene fusion event (the two truncated reads from the same fusion event should have the same ID).
 6. It is difficult to conclude that “overall, the SV detection with short read WGS was useful except for the detection of translocations” only based on the proportion of short-reads based SVs covered by long reads WGS (lines 224-225 on page 8). Sequencing depth matters since these are somatic SVs. As stated by the authors, the average sequencing depth of short reads tumor WGS is 97x while the depth is 24x for long reads. The proportion of SVs not covered by long reads WGS is probably due to the lower depth, so this proportion is underestimated. The more precise statement might be something like “... translocations detected by short reads WGS is less reliable since fewer of them are covered by long reads WGS compared with other SVs...”.
 7. Overall tumor DNA methylation levels are compared among AIS, MIA/Lepidic-Ad, and Advanced-Ad samples. The authors’ conclusion is “Hypomethylation significantly occurred in MIA cases while the genomes of AIS cases will retain a normal-level high methylation status”(lines 297-299 on page 10). I argue that this should be done by using the difference in methylation level between tumor and normal samples of each individual rather than comparing tumor methylation levels as well as the median methylation level of normal samples. Since methylation level is strongly associated with many factors such as age even for normal samples, there are many known and unknown confounding factors if tumor methylation levels are compared directly.
 8. The authors reported copy number loss events are associated with the degree of hypomethylation status (lines 302-303 on page 10). The evidence is the scatter plot in Fig 5d. Some quantitative criteria such as the correlation coefficient should be provided for a solid conclusion.
 9. The authors stated that the order of mutation occurrence can be resolved by integrating phasing information provided by long reads (lines 394-404 on page 13). However, the method is not

explained explicitly (not found in the Methods section either). I guess that if two somatic mutations are covered/phased by the same long read then the one with higher VAF occurred first. The theory/assumption behind this and exact steps to perform the analysis should be explained.

Minor points:

1. Fig 4a in line 391 page 13 is not relevant to the text.
2. The colors for “VAF<0.2” and “VAF>=0.2” in Fig 7a are hard to distinguish.
3. It is recommended to perform several rounds of proofreading to avoid misleading mistakes such as “S11 and S14” on line 314 page 10.
4. This is not a critical comment but just out of curiosity, is it helpful to add haplotype information to PyClone? like, using haplotype-specific somatic mutations as the input.

Point-by-point responses to Reviewer #1:

Reviewer #1 (Remarks to the Author): Expert in genomics, bioinformatics, and long-read sequencing

The authors performed multi-omics experiments, including short-read WGS, long-read WGS, short-read RNA-seq, and spatial transcriptome sequencing on 76 lung cancer samples (48 early-stage cases, 20 advanced-stage cases, and 8 others) and their matched normal samples. By performing analyses including profiling somatic SNV, CNV, SV, haplotype-specific SNV/SV, DNA methylation, mutation abundance, mutational signatures, gene expression, and clonal architecture reconstruction, the major conclusions reported by authors include 1) the driver mutations already occur in AIS, whereas other cancerous aberrations majorly appear at later stages; 2) significant changes in environmental factors impacted AIS Noguchi type B, followed by dramatic changes in genomic aberrations in the invasive stage of lung adenocarcinoma. 3) there were different molecular features between Noguchi type A and type B in AIS at the genome and transcriptome levels.

I have the following comments regarding the methods, results, and academic writing:

Major comments:

1. How the driver mutations are obtained is not clearly stated. Although EGFR mutation is apparently obtained from previous publications (10, 11, and 15), other driver mutations such as MET, KRAS, etc need sources (lines 174-176 on page 7). To make the claims solid, it is required to add references or other evidence to both the main text and Table S1, where driver mutations are listed.

Thank you for pointing this out. We added further references on the definition of driver mutations to the main text (lines 174–177). The drivers were selected from:

1. The Cancer Genome Atlas Research Network. Comprehensive molecular profiling of lung adenocarcinoma. 2014 *Nature*.
2. Ding L *et al*. Somatic mutations affect key pathways in lung adenocarcinoma. 2008 *Nature*.
3. Saito M *et al*. Gene aberrations for precision medicine against lung adenocarcinoma.

2016 *Cancer Sci*.

4. Dogan S *et al*. Molecular epidemiology of EGFR and KRAS mutations in 3,026 lung adenocarcinomas: higher susceptibility of women to smoking-related KRAS-mutant cancers. 2012 *Clin Cancer Res*.
5. Tate JG *et al*. COSMIC: the Catalogue Of Somatic Mutations In Cancer. 2019 *Nucleic Acids Res*.
6. Ou SHI *et al*. HER2 Transmembrane Domain (TMD) mutations (V659/G660) that stabilize homo- and heterodimerization are rare oncogenic drivers in lung adenocarcinoma that respond to afatinib. 2017 *J Thorac Oncol*
7. Sheikine Y *et al*. BRAF in Lung cancers: analysis of patient cases reveals recurrent BRAF mutations, fusions, kinase duplications, and concurrent alterations. 2018 *JCO Precis Oncol*.
8. Arcila ME *et al*. MAP2K1 (MEK1) mutations define a distinct subset of lung adenocarcinoma associated with smoking. 2015 *Clin Cancer Res*.

We especially listed previously reported driver mutations in oncogenes (refs 1–3). We believe that those mutations are the most representative, in addition to *EGFR* mutations. For example, *KRAS* G12 mutations are well-known hotspot mutations (refs 4) and *ALK/RET* fusions and *MET* exon 14 mutations have been previously reported as driver aberrations (refs 1 and 3). For other driver genes, such as *ERBB2*, *BRAF*, and *MAP2K1*, we checked the COSMIC database (ref 5) and various articles (refs 6–8) for confirming their oncogenic functions. This description is provided in the footnote of **Supplementary Table S1**.

2. The somatic mutations are apparently obtained by comparing a tumor sample and the matched “normal” sample from the same individual. However, details about how the “normal” samples are defined and obtained are not stated in the manuscript. This is critical for reproducibility. It is also recommended to clearly state that “mutation” in this work means “somatic mutation” to avoid misunderstanding for the general audience.

Thank you for the comment. We added the following description of normal samples in the **Methods** section (lines 500–505):

“Frozen surgical specimens from lung cancer patients were used in this study. Both

tumor tissue and normal counterpart (peripheral lung) tissue were obtained separately from the same resected specimen. Normal counterpart tissue was carefully collected from the regions >5 cm from tumors with macroscopically normal morphology. The pathologists (HK and MN) confirmed using light microscopy that the normal counterpart tissue did not contain tumor cells.”

According to the reviewer’s recommendation, we also exchanged the word “mutation” by “somatic mutation” in the revised manuscript.

3. The claim that “AIS cases are still free from mutations in the tumor-suppressor genes, especially TP53” lacks statistical tests (lines 185-186 on page 7). One of the evidence supporting this conclusion is 7 out of 48 Early-Ad cases harbored relevant mutations in tumor-suppressor genes while 11 out of 28 (28 = 76 - 48) Advanced-Ad cases “harbored deleterious mutations in tumor-suppressor genes”. A statistical test is needed since the sample size is not very large. I did a quick Fisher Exact Test and the p-value is about 0.024, which is barely significant. The mutations on tumor-suppressor genes in Advanced-Ad are not “far more” frequent than that in Early-Ad cases as claimed by the authors if randomness is considered. The other evidence is Fig 1c, which provides gene-by-gene mutation frequencies. Adding a statistical test for each gene is required for a solid conclusion.

The R (version 4.2.2) code for Fisher Exact Test is:

```
dat <- matrix(c(7, 48 - 7, 11, 28 - 11), nrow = 2, dimnames = list(c("mutation present",  
"mutation absent"), c("Early-Ad", "Advanced-Ad")))  
fisher.test(dat, conf.int = FALSE)
```

We appreciate this comment. First, please allow us to point out that all indicated numbers by the reviewer should not be correct. Here, we attempted to examine only adenocarcinoma cases, thus excluding five non-adenocarcinoma cases. Therefore, tumor-suppressor genes were mutated in 7 and 11 cases in Early-Ad (48 cases) and Advanced-Ad (23 cases) groups, respectively. Using these values, we performed Fisher’s exact test as indicated by the reviewer. As a result, the p-value was 0.0072 and significant. We added this result to the revised manuscript (**lines 183–184**). Nonetheless, we agree that the sample size is not always large. We added this caveat to the main text and edited the

term “far more frequent” to just “more frequent.”

We also performed a similar statistical test for *EGFR*, *TP53*, *SMARCA4*, and *RBM10* in adenocarcinoma cases (48 Early-Ad + 23 Advanced-Ad cases). We confirmed that mutations in *TP53* were more frequently found in Advanced-Ad cases ($p = 0.0012$). Similarly, *SMARCA4* was more frequently mutated in the Advanced-Ad group ($p = 0.035$). On the other hand, the p-value for *RBM10* mutations was not statistically significant ($p = 0.25$) given the small total number of mutated cases. We included these results and discussion in **Figure 1c** and the revised text (**lines 186–187, 204, 207–209**).

4. I am not convinced by the claim “Exceptionally, long-range deletion and copy number loss events of CDKN2A/p16 region were identified even in Early-Ad cases (Fig. 1f and Supplementary Fig. S1), which is consistent with a previous report, collectively suggesting that the key driver gene mutations and this gene deletion should be essential to initiate the early cancer.” (lines 192-196 on page 7) for three reasons:

- 1) Logically, a mutation (SNV + CNV, etc) event present in the early stage does not prove that it is essential to initiate early cancer. Evidence that the absence of this mutation inhibits the initiation of early cancer is needed to make this claim.**
- 2) Fig 1f does not include any information about “long-range deletion and copy number loss events of CDKN2A/p16 region”.**
- 3) It is not clear that there is a deletion in the CDKN2A region based on the IGV snapshot of Supplementary Fig. S1a. In addition, the color code in the figure is not explained.**

Thank you for your careful comment. To describe *CDKN2A*, we modified the revised text as below;

- 1) We did not mean that *CDKN2A* mutations are essential to initiate cancers. In fact, as the reviewer pointed, *CDKN2A* mutations were not always found in Early-Ad cases. Please note that *CDKN2A* mutations were identified in the region affected by gene deletions or copy number loss, even though those regions are not commonly detected in Early-Ad cases; therefore, these occurrences were considered “exceptional.” Nevertheless, a previous study (*Clin Cancer Res* 2008) indicated that, when present, *CDKN2A* loss events occur already from very early stages, unlike the other tumor-suppressor mutations.

- 2) We apologize for the unclear description. “Long-range deletion” in **Figure 1f** means “Gene deletion.” We modified the figure and the main text accordingly and added representative cases of gene deletions to **Supplementary Figure S2**.
- 3) In the revised supplementary information (**Supplementary Figs. S2a–d**), we clarified the location of mutant reads and clarified the description in the legend where we added the color code.

We changed the sentence to “*In accordance with previous reports, gene deletion and/or copy number loss events of CDKN2A were identified in several cases; they were also observed in Early-Ad stages, although the disruption of tumor-suppressor genes rarely occur at these stages (Fig. 1f and Supplementary Fig. S2). These results suggested that key driver gene mutations initiate the development of early stages of cancer, followed by the deletion of this gene, which might promote cancer cell proliferation during early progression of adenocarcinomas.*” (lines 193–199).

5. In line 218 of page 8, “... driver fusion KIF5B-RET...” while in Fig 2b it is “KIF5B-RET inversion”. The difference should be explained or corrected if it is a typo. In addition, the IDs/names of the truncated reads in KIF5B and RET region should be provided to prove there is a gene fusion event (the two truncated reads from the same fusion event should have the same ID).

We apologize the confusing wording. These two words indicate the same phenomenon. This inversion event of the genomic regions comprising *KIF5B* and *RET* resulted in *KIF5B-RET* gene fusion. To clarify this point, we edited “*KIF5B-RET* inversion” to “*KIF5B-RET* fusion” in **Figure 2b**.

We extracted three truncated reads from *KIF5B* and *RET* by nanomonsv. We re-visualized both primary and supplementary alignments on IGV and confirmed that the truncated reads were split-mapped to both *KIF5B* and *RET*. Thus, we provide the name of the truncated reads in **Figure 2b**. Similarly, we performed the same analyses for the *MET* deletion.

6. It is difficult to conclude that “overall, the SV detection with short read WGS was

useful except for the detection of translocations” only based on the proportion of short-reads based SVs covered by long reads WGS (lines 224-225 on page 8). Sequencing depth matters since these are somatic SVs. As stated by the authors, the average sequencing depth of short reads tumor WGS is 97x while the depth is 24x for long reads. The proportion of SVs not covered by long reads WGS is probably due to the lower depth, so this proportion is underestimated. The more precise statement might be something like “... translocations detected by short reads WGS is less reliable since fewer of them are covered by long reads WGS compared with other SVs....”.

Thank you for your comment, with which we agree. In fact, the translocation should be the most difficult target for SV detection by short reads. We observed that, at a given sequencing depth, detection of other types of SVs, such as deletions, inversions, and duplications, was relatively precise. Accordingly, we modified this sentence as shown below (lines 229–231);

“Translocations detected using short read WGS were relatively less reliable, as few translocations were covered by long read WGS compared with other SVs.”

7. Overall tumor DNA methylation levels are compared among AIS, MIA/Lepidic-Ad, and Advanced-Ad samples. The authors’ conclusion is “Hypomethylation significantly occurred in MIA cases while the genomes of AIS cases will retain a normal-level high methylation status”(lines 297-299 on page 10). I argue that this should be done by using the difference in methylation level between tumor and normal samples of each individual rather than comparing tumor methylation levels as well as the median methylation level of normal samples. Since methylation level is strongly associated with many factors such as age even for normal samples, there are many known and unknown confounding factors if tumor methylation levels are compared directly.

Thank you for this important suggestion. According to your suggestion, we calculated the difference in methylation levels between tumor and normal counterparts and compared them with AIS, MIA/Lepidic-Ad, and Advanced-Ad cases for each individual, separately. We confirmed that the results were almost identical. Hypomethylation was even more significantly detected in MIA/Lepidic-Ad cases. To show this result, we added

the median methylation rates of the normal counterparts of each case to **Figure 5a** in the revised manuscript. In addition, note that, when we examined normal DNA methylation levels (**Supplementary Fig. S6b**), we similar methylation levels among individuals.

8. The authors reported copy number loss events are associated with the degree of hypomethylation status (lines 302-303 on page 10). The evidence is the scatter plot in Fig 5d. Some quantitative criteria such as the correlation coefficient should be provided for a solid conclusion.

According to the reviewer's comments, we calculated the Spearman correlation coefficient between DNA methylation rates and the number of somatic mutations. To address comment #7, we used differences in DNA methylation levels between tumor and normal samples for this analysis. As a result, in addition to the copy number loss, we found that the total copy number changes and the number of insertions show a moderate but statistically relevant correlation ($|r_s| = 0.46-0.56$; $p = 5.3e-6-0.00032$). We added the new plots and Spearman's correlation coefficients to **Figure 5d** and modified the revised manuscript (**lines 308-312**) as shown below:

“We also found that a number of large insertions and copy number changes (especially copy number loss events) were moderately associated with the degree of hypomethylation (Fig. 5d). The time when genome-wide hypomethylation is MIA/Lepidic-Ad or later stages, in which initiate the increasing of genome instability levels.”

9. The authors stated that the order of mutation occurrence can be resolved by integrating phasing information provided by long reads (lines 394-404 on page 13). However, the method is not explained explicitly (not found in the Methods section either). I guess that if two somatic mutations are covered/phased by the same long read then the one with higher VAF occurred first. The theory/assumption behind the this and exact steps to perform the analysis should be explained.

We apologize for the missing explanation. We employed a simpler or more direct method.

We assumed that if long reads harbor mutations A and B and long reads harbor only mutation A, the order should be A -> B. Practically, the presence of mutations A and B was considered as VAFs. We clarified this method in the **Methods** section (**lines 758–768**) as below:

“Extraction of the order of mutation occurrences

For pairs of two somatic mutations, their occurrence order was resolved using long reads which directly covered both positions of the two mutations. If a mutation with a lower VAF was fully detected with the other mutation with a higher VAF in the same long reads (double-mutant reads), the mutation with a higher VAF occurred first, and these mutation pairs were assigned to the same haplotype.

To perform this analysis, we evaluated whether the detected somatic mutations from the short read WGS existed in haplotype-tagged long reads using SAMtools (version 1.12) mpileup. For determining the order in each mutation pair, ≥ 3 single-mutant reads where only the mutation with a higher VAF was detected and ≥ 3 double-mutant reads without inconsistencies needed to be extracted.”

Minor points:

1. Fig 4a in line 391 page 13 is not relevant to the text.

We corrected the text by removing the figure number.

2. The colors for “VAF<0.2” and “VAF>=0.2” in Fig 7a are hard to distinguish.

We added the fill pattern to the bar graph and changed the color plot of **Figure 7a**.

3. It is recommended to perform several rounds of proofreading to avoid misleading mistakes such as “S11 and S14” on line 314 page 10.

We apologize for the unclear explanation. We edited “S11 and S14” to “case S11 and case S14.”

4. This is not a critical comment but just out of curiosity, is it helpful to add haplotype information to PyClone? like, using haplotype-specific somatic mutations as the input.

Thank you for the suggestion. We agree that haplotype information should be potentially helpful for the PyClone analysis. In fact, we applied such an analysis but the obtained results were not sufficiently precise. This may be because the constructed haplotype blocks are still fragmented and aneuploidy not properly considered. To address this concern, improved long-read sequencing is needed; We will tackle this issue in future work.

Point-by-point responses to Reviewer #2:

Reviewer #2 (Remarks to the Author): Expert in spatial transcriptomics, cancer transcriptomics, and tumour microenvironment

The manuscript describes the results of whole-genome sequencing analysis of 76 lung cancer specimens, including 48 early and small-sized lung adenocarcinoma cases and 8 cases of advanced adenocarcinomas. The sequencing data were analyzed to detect somatic mutations, including point mutations, SVs, and CNVs. The analysis revealed that EGFR driver mutations were acquired early in 63% of the Early-Ad cases, and the frequencies of EGFR mutations among the Early-Ad cases were similar between the different pathological classes and the advanced cases. The driver compositions observed were broadly similar to those observed in advanced adenocarcinomas, suggesting that the most important key driver mutations occur at early stages.

This manuscript offers an important addition of a novel cohort of early-stage lung cancer lesions to the existing large databases of lung adenocarcinoma of the TCGA and ICGC. It could benefit from a more comprehensive discussion of clinical & diagnostic implications. I found the biological contextualization to be underwhelming.

The rich resource and the wealth of performed molecular assays is important for the community, but reuse of the dataset will require facilitated access to the repository, availability of processed datasets, and the upload of well-annotated analysis code.

Thank you for indicating the importance of this manuscript. We prepared the requested datasets and analytical codes and shared them.

Major comments:

Data availability: I was not able to assess the completeness of the generated datasets due to the controlled-access criteria of the used repository. The authors report in their “reviewer’s token file” that all the raw data has been made publicly and freely available. I was not able to complete the lengthy NBDC application within the scope of this review and would appreciate an actual reviewer’s token for easier access to the raw data. Additionally, I expect that the authors also provide the final processed files to facilitate

the reproduction of the reported results. Processed data (excluding raw sequencing reads) could be uploaded to an open-access repository.

For the raw sequencing data, we already made publicly available the whole dataset from the JGA of NBDC. However, NBDC/JGA does not allow anonymous reviewers to access the controlled access data because of ethical guidelines in Japan. To confirm the completeness of the generated datasets, please check the dataset description in the NBDC webpage (<https://humandbs.biosciencedbc.jp/en/hum0068-v6>). We apologize for not being able to change NBDC's policy.

For the processed data, we prepared the lists of somatic mutations, DNA methylation profiles, gene expression levels, and spatial omics profiles including Visium, PhenoCycler, Xenium, and all the relevant datasets and uploaded the files in our open-access database DBKERO (Suzuki A *et al.* 2018 *Nucleic Acids Res*). We added the URL of the download page to the **Data Availability** section in the revised manuscript (**lines 779–781**) as below;

“The processed data was deposited in the database DBKERO (<https://kero.hgc.jp/>) and made publicly available in the download page (https://kero.hgc.jp/Early_cancer.html).”

Reproducibility of computational analyses: The paper relies on extensive computational analyses, yet the provided computational methodologies are superficial and do not enable independent reproduction of the analyses. For reproducibility purposes, the authors need to provide access to the underlying code in an appropriate repository (ideally with DOI, eg Github+Zenodo).

We provided our scripts for DNA methylation analyses, transcriptome analyses, haplotype assignment, and phasing of somatic mutations in GitHub. We added the **Code Availability** section to the revised manuscript and provided the detail information (**lines 784–786**) as below:

“Code Availability

The code used in this study is available in the GitHub repository at https://github.com/asuzuki-asuzuki/Early-Ad_2023.”

Spatial transcriptomic analyses: the loupe browser is suitable for exploratory visualizations but not for reproducible analyses and for the generation of publication-quality figures. The authors should use one of the many suitable coded analysis pipelines for visium analysis to ensure reproducibility.

Thank you for your constructive suggestion. By using the R package Seurat, we re-analyzed Visium datasets and re-generated the figures (**Fig. 6** and **Supplementary Fig. S8**). That way, we confirmed that the results coincide.

The reporting of the transcriptomic analyses seems incomplete. The authors jump straight to a DGE of Noguchi type A vs type B and report the enriched GO terms. It would be of interest to see a PCA plot of all the generated transcriptomic datasets to better understand the landscape of the assessed samples.

According to the reviewer's suggestion, we performed PCA using the expression levels of 11,027 genes (with >5 rpkm in ≥ 1 case) for the 12 Early-Ad and six Advanced-Ad cases. Some AIS Noguchi type B cases showed a distinctive landscape even in whole transcriptome status, indicating that the transcriptome signatures of Noguchi type B tumors might differ from those of Noguchi type A tumors due to the beginning of alveolar collapse and their interaction with the microenvironment. We included the plot in **Supplementary Figures S7a** and **S7b** and added the corresponding discussion to the legend.

Bulk RNA samples might be confounded by the presence or lack of surgical margins, infiltration of other cell types and handling time after surgery. The authors use Visium to validate bulk RNA sequencing. Neither of these methods enable single cell resolution. Some of the changes are then validated with imaging data (single cell resolution). Have the authors considered computational deconvolution or mapping to existing single cell RNA sequencing datasets for this disease entity?

We totally agree with your comment –and a similar one from reviewer #3– that Visium is not a technique at single-cell resolution. To address this issue, we conducted extensive analyses of bulk RNA-seq and Visium data as follows:

- 1) Bulk RNA-seq deconvolution: we conducted deconvolution analysis using CIBERSORTx for the Early-Ad bulk RNA-seq datasets using single-cell RNA-seq data of lung adenocarcinoma (Zhu J *et al.* 2022 *Exp Mol Med*) (**Supplementary Figs. S7c–S7e**). As a result, various types of immune cells were infiltrated; in addition, we confirmed that some Noguchi type B stage cases had an increased proportion of cytotoxic immune cells including CD8+ T cells, NK and NKT cells increased.
- 2) Visium deconvolution: We performed computational deconvolution analysis of Visium data of case TSU-21 using the single-cell RNA-seq data (Zhu J *et al.* 2022 *Exp Mol Med*) to characterize local immune cell profiles (**Supplementary Figs. S8d and S8e**). Interestingly, we found several different type of macrophages (maybe alveolar macrophages). In particular, FABP4+ macrophages and SPP1+ macrophages (a subpopulation of CCL3L1+ macrophages) were found in certain alveoli structures. The most highly *FABP4* and *SPP1* expressed macrophages were enriched near the collapsed alveolar region.
- 3) Xenium validation: To validate the status of tumor cells and their microenvironment at a single-cell level, we conducted *in situ* gene expression analysis using Xenium (10x Genomics) in the following section, which enables spatial expression profiles at a single-cell/subcellular level. We obtained Xenium data of case TSU-21 using a fresh frozen tissue section from the same tissue block used for Visium analysis (**Fig. 6i, Supplementary Fig. S10, Supplementary Tables S10, and S11**). Using Xenium data, we also performed deconvolution analysis of Visium data (**Supplementary Fig. S10**). Variation in macrophage subsets (i.e. SPP1+, FABP4+ and CXCL9+) was also validated at the single-cell level (**Fig. 6j and Supplementary Fig. S10**).

Thus, through these extensive analyses, we confirmed our claims. We included this discussion to the text (**lines 339–343 and 379–396**).

Minor:

Increase quality/resolution of the plot and text in Figure 7e. Currently, the legend is not legible upon enlargement.

We increased plot resolution and provided additional text in **Figure 7e** in the revised manuscript.

Increase quality/resolution of Visium plots in Supplementary Figure 6. Currently, the legend is not legible upon enlargement.

We increased resolution of the Visium plots (**Supplementary Fig. S8** in the revised manuscript) and made the Visium plot using Seurat instead of Loupe Browser.

Figure 8c right plot: background plot seems to be a low-quality screenshot, please update.

We updated the right plot of **Figure 8c** in the revised manuscript.

Point-by-point responses to Reviewer #3:

Reviewer #3 (Remarks to the Author): Expert in lung cancer genomics and evolution

Thanks for the opportunity reviewing this interesting study by Haga etc. on the molecular evolution during early carcinogenesis. This is the first study using long read WGS and spatial gene expression analyses on this topic that I am aware of. Therefore, the data is welcomed to the field. However, the study still has some important limitations as listed in the following.

1. The sample size is still pretty small. It is understandable that these lesions may not be as many as invasive diseases. The authors should discuss this important limitation and the results should be interpreted with caution.

Thank you for indicating this important point, with which we fully agree. In fact, we know that we did not have a large number of samples in this study due to clinical reasons. Firstly, the number of surgical operation cases is smaller than advanced cancers. Secondly, for long read sequencing, we need to use fresh frozen tissues of very early cancer stages, which are inherently far smaller than advanced cancers, which made it sometimes difficult to extract sufficient amounts of DNA from those small specimens. Despite the limited number of samples, we could characterize important features of early cancers as supported by statistical analyses (**Fig. 1c** and **Fig. 5d**). We added the following description to the text (**lines 474–485**):

“It is true that the number of samples collected for this study was not large. This was for the following clinical reasons. First, the number of patients who underwent surgery was relatively low: i.e. patients with advanced cancers were more likely to undergo surgery. Second, for the long read sequencing, it was necessary to extract DNA from fresh frozen AIS and MIA tissue specimens less than 2 cm in diameter and these were lepidic-type adenocarcinomas histologically, containing a smaller number of tumor cells, than advanced adenocarcinomas, most of which were papillary, acinar or solid-type adenocarcinomas. Therefore, it was not always easy to extract sufficient amounts of DNA from AIS and MIA specimens. However, even with this limited number of samples, we were able to characterize important features of early cancers as described here. Note that the described differences are supported by statistical tests (Fig. 1c and Fig. 5d).”

2. The authors used multiomics approaches, which is an important strength of this study. However, it is not easy to follow which samples had short read, long read, Visium, bulk seq, mIF etc. It will be helpful to have a schema of the study to illustrate the information and the rationale for certain assays. For example, how many samples went for Visium? How did you decide to choose samples for Vixium etc.

We added the study schema and the sample size for each analysis in **Supplementary Figure S1**. Detailed information on samples and assays was included in the legend of **Supplementary Figure S1** as below:

*“Fresh frozen tissues from surgical specimens of 76 lung cancer cases were used for this study. Short read WGS sequencing was performed for tumor-normal tissue pairs of all 76 cases. Long read sequencing was performed using high-molecular weighted DNA samples from 62 tumor and 59 normal tissues, respectively. For validation of somatic mutation detection, target captured sequencing was conducted for 42 tumor samples using the remaining DNA samples. For transcriptome analysis, RNA-seq (poly-A method) was performed for 18 tumor samples whose RNA samples were not highly degraded. Using OCT-embedded frozen tissues, spatial omics analysis was conducted for analysis of gene expression patterns considering histological and spatial information. Spatial transcriptome sequencing analysis (Visium) and multiplexed fluorescence immunostaining (PhenoCycler) were performed for three representative Early-Ad cases. These cases harbor typical histological characters in AIS Noguchi type A, AIS Noguchi type B, and MIA Noguchi type C. For further validation of spatial expression patterns at the subcellular level, in situ gene expression profiling (Xenium) was performed for one AIS Noguchi type B case using a tissue section nearby those used for Visium/PhenoCycler analysis. *Twenty datasets previously analyzed were included in this study as well.”*

3. Most findings have been reported in the previous studies, therefore, these results are confirmatory. I would suggest the authors put their cards down and emphasize the novel findings in this study in the abstract and in the summary/conclusions.

Thank you for the suggestions. In fact, some results were previously reported and others are novel. Particularly, we found that Noguchi type B tumors should be at the most crucial stage when drastic changes are triggered in various multi-omics layers. This part is further validated by the additional data of the deconvolution and the Xenium analyses. For other points, we clearly indicated the unique findings in this study in the abstract and conclusions. We modified the descriptions for which the obtained results are confirmatory to the previous studies.

4. The association between EGFR mutations and TMB, CNV burden etc. is interesting. Did the authors control for smoking status? One plausible explanation the authors presented is the association of EGFR mutations with DNA repairs. Can authors analyze their own data, mutations or gene expression-based pathway analysis between EGFR WT vs EGFR mutant to provide the evidence to support this?

Thank you for this comment. In the previous version of the manuscript, we did not include smoking status in this analysis. We compared the mutation burden between smokers and non-smokers. As a result, *EGFR* mutation status and mutation burden showed a significant association especially in non-smokers among the Early-Ad cases for all mutation types (**Supplementary Fig. S4d**). We confirmed that this association was not affected by mutation accumulation due to tobacco smoking mutagens.

To examine the association with DNA repair pathways, we extracted differentially expressed genes between *EGFR*-mutant and wild-type cases and performed GO enrichment analysis. However, no DNA repair-related pathways appear at a global level. We could not precisely identify which DNA repair-associated molecule or, perhaps, a sub-network may be selectively disordered. Therefore, we could not confirm the contribution of *EGFR* mutations to this difference in mutation burden. We included this discussion in the revised text.

5. The authors have generated a lot of interesting data, which can be dug in deeper. For example, deconvolution can be applied to RNA seq data to look into the immune

infiltration by TIMER, CIBERSORT, etc. How many cases underwent Visium profiling? These spatial gene expression data can be explored in a much deeper way such as difference between different histological subtypes; between tumor, tumor margins, distant normal lung; cell-cell interaction; ligand-receptor analyses etc.

Thank you for your constructive comments. Reviewer #2 also pointed out this issue, which we addressed in our response to their major comment #5. We refer you to that point, therefore.

To address your other concerns, we examined the patterns of immune infiltration using bulk RNA-seq data. We conducted deconvolution analysis using CIBERSORTx for the Early-Ad bulk RNA-seq datasets using single-cell RNA-seq data of early lung adenocarcinoma (Zhu J *et al.* 2022 *Exp Mol Med*) (**Supplementary Figs. S7c–S7e**). As a result, various types of immune cells were infiltrated; in some cases, there was an increased proportion of cytotoxic immune cells including CD8+ T cells, NK, and NKT cells at the Noguchi type B stage (**lines 339–343**).

For our Visium analyses, we analyzed three types of Early-Ad cases (AIS Noguchi type A, AIS Noguchi type B, and MIA Noguchi type C). For additional interpretation, we further performed clustering analysis and extracted marker genes to characterize each local region (**Supplementary Fig. S8c**). Early-Ad tumors often exhibit pure lepidic growth. However, in Noguchi type B tumors (case TSU-21), four tumor cell clusters were identified. We found that the *FOS/JUN*-highly-expressed cluster overlapped with the region of alveolar collapse. We added *FOS*, *FOSB* and *JUN* expression patterns of case TSU-21 to the revised text (**Fig. 6h, lines 374–375**).

Similarly, we conducted deconvolution analysis of Visium data using scRNA-seq data (Zhu J *et al.* 2022 *Exp Mol Med*) (**Supplementary Figs. S8d and S8e**). We found that several different types of macrophages were distributed in the Noguchi type B tumor of case TSU-21. In particular, FABP4+ and SPP1+ macrophages were localized in certain alveoli structures, and macrophages of which FABP4 or SPP1 highly expressed were enriched near the region of alveolar collapse. We also found such features in Xenium data of the same case (**Supplementary Figs. S10d–f**). Thus, we could characterize the features of the Noguchi type B region based on both tumor and immune cells.

We also conducted a thorough analysis of the Visium data regarding ligand-

receptor analyses using CellChat (**Supplementary Fig. S8g**). We found abundant interactions between macrophages (cluster 1) and fibroblasts (cluster 7) and identified 185 integrations among these clusters ($p < 0.05$). As an example, a result of the SPP1 signaling network is shown in **Supplementary Figure S8h**. SPP1 might be secreted from macrophages and received by integrin complexes in fibroblasts (**lines 387–390**).

6. The immune features during early lung cancer development are important. There are recent publications in this space. I would suggest the authors to dig more by including more mIF data or at the minimum deconvolute the RNA seq data to address this important aspect and discuss their findings in the setting of recent publications on this topic. There are two publications at least on Nature Communications by Chen et al in 2019, PMID: 31784532 and the other one by Dejima et al PMID: 33976164.

We appreciate the reviewer's suggestions. We conducted further analyses to characterize tumor infiltrating immune features. As in our response to comment 5, we performed deconvolution analysis of bulk RNA-seq data. As a result, various types of immune cells were infiltrated; in some cases, there was an increased proportion of cytotoxic immune cells including CD8+ T cells, NK, and NKT cells at the Noguchi type B stage (**Supplementary Figs. S7d and S7e**). Next, to compare our data with recent publications indicated by the reviewer, we evaluated the association between the immune infiltration levels and several omics features, such as TMB, CNV burden, and DNA methylation levels. We found that TMB showed a slight, positive correlation with the infiltration levels of cytotoxic cells while the number of CNV events was not significantly correlated (**Supplementary Fig. S7f**).

Chen *et al.* (Chen *et al.* 2019 *Nat Commun*) indicated that HLA LOH and chromosome 6p loss/gain is associated with immune features. We observed only two Early-Ad cases and six advanced cases harboring 6p loss/gains. This may explain why we could not find a significant association between 6p CNV and immune infiltration. However, we consider that sequencing data of variable HLA loci may be fairly evaluated. We would like to examine this issue in the future.

Moreover, we found that genome-wide DNA hypomethylation was associated with increased CD4+ Treg and decreased cytotoxic immune cells (**Supplementary Fig.**

S7g). This indicates that global hypomethylation is associated with immune suppression, which is consistent with previous results (Dejima *et al.* 2021 *Nat Commun*).

All these results were added to the revised text (**lines 339–343**) and discussed in the legend of **Supplementary Figure S7**.

7. The authors can do a better job integrate these multiomics data.

We appreciate the reviewer's suggestions. As mentioned above, we have now integrated the multi-layered datasets, especially for the elucidation of the association among genomic/epigenomic features and immune features (e.g., **Fig. 5d** and **Supplementary Figs. S7f** and **S7g**). In addition, we further attempted to analyze and integrate genomic and epigenomic layers from a yet different viewpoint. Through this extensive analysis, we attempted to demonstrate the power of our multi-omics analysis.

We first focused on DNA methylation statuses of cancer-related genes in which genomic aberrations were already reported in **Figure 1b**. We calculated the rates of DNA methylation at the promoter region of those cancer-related genes (**Supplementary Fig. S6e**). We confirmed that the driver gene in each case showed a low methylation rate. On the other hand, for tumor-suppressor genes, hypermethylation occurred at the promoter region. Hypermethylation was observed in the *CDKN2A* and *APC* gene promoters despite the lack of genomic mutations. Interestingly, the downstream alternative promoter region of *APC* was highly methylated in several Early-Ad cases as indicated in the revised text (**line 325–326**).

We further refined this multi-layered characterization to the haplotype level analysis. We observed distinct DNA methylation rates between haplotype #1 and #2 at the *APC* promoter region in some cases (**Supplementary Fig. S6f**). In an Early-Ad case (case TSU-31, MIA), one haplotype was highly methylated but the other remained lowly methylated (**Supplementary Fig. S6g**), indicating that the function of this gene should be retained. On the other hand, in an advanced case (case S14), hypermethylation occurred in one haplotype and LOH occurred in another, resulting in functional loss of the gene (**Supplementary Fig. S6h**). This gene should have started to be disrupted firstly due to epigenomic aberrations and then due to genomic mutations during the progression of adenocarcinoma. We added this discussion to the legend of **Supplementary**

Figure S6.

8. Finally, the term evolution in the title and body of the manuscript may not be appropriate. Although these may be different stages during lung cancer evolution, all these specimens are resected specimens. They are independent to each other and cannot be used to infer the evolutionary process as they are not related. All observations are association that may suggest the evolutionary relationship. This is a universal issue in the field of cancer evolution research. But the authors should discuss this limitation.

Thank you for your comment, with which we completely agree. As the samples are independent, we could not directly trace the evolutionary process from very early to advanced stages for each case. We could just infer the history of cancer evolution by PyClone analysis and mutation phasing analysis. Nevertheless, we consider that the real evolutionary aspects of cancers should be represented by the sum of individual cancers, at least, to some extent. However, we added this caveat to the revised manuscript (**lines 465–469**):

“As the samples were independent of each other, we could not directly trace the evolutionary process from very early to advanced stages for each case. We could only infer the general history of cancer evolution by PyClone analysis and mutation phasing analysis. Nevertheless, we consider that the real evolutionary aspects of cancers should be represented by the sum of individual cancers, at least, to some extent.”

REVIEWER COMMENTS

Reviewer #1 (Remarks to the Author):

The authors have addressed all my comments.

Reviewer #2 (Remarks to the Author):

The authors have revised their scientific manuscript, providing thorough responses to each of the reviewer's comments. The authors addressed my concerns regarding data availability by making the raw sequencing data publicly available in the Japanese Genotype-phenotype Archive of the National Bioscience Database Center (NBDC/JGA), which they acknowledge has a restricted access due to ethical guidelines in Japan. Furthermore, they have deposited the processed data, such as lists of somatic mutations, DNA methylation profiles, and gene expression levels, in an open-access database. Regarding the computational analyses reproducibility, the authors have made their scripts publicly available on GitHub, allowing for the replication of their analyses.

In response to the reviewer's comment on potential confounding variables in bulk RNA samples, the authors used deconvolution analyses of bulk RNA-seq and Visium data. They compared these to single-cell RNA-seq data of lung adenocarcinoma, which identified various infiltrating immune cell types and subtypes of macrophages. They further validated these findings at a single-cell level using Xenium data.

I very much appreciate the addition of the Xenium dataset to this revised manuscript. It strengthens the conclusions regarding the TME composition. However, the current state of analysis seems a bit premature in the revised manuscript.

Specific comments:

The manuscript compares gene expression features between Noguchi type A and B samples. The analysis with the custom xenium lung panel was only performed with one Noguchi type B sample. Could the authors also analyze a Noguchi type A sample for comparative reasons?

I appreciate that the authors list the obtained transcript counts per cell in 6i. Could the authors comment in the manuscript or methods and contextualize this capture rate / the corresponding RNA quality of the sample and what one would expect based on the information supplied by the manufacturer or the literature?

The TME composition in 6j is illustrated with a few marker genes of interest, could this analysis also be performed with the detected clusters in 6i (unsupervised)?

The authors mention macrophage localization patterns in 6j, it would be nice to have quantitative measures of this localization pattern and statistical test (neighborhood analysis, distance..) that compare it with a control sample. The current phrasing "FABP4+ and SPP1+ macrophages were localized in certain alveoli structures" is not specific enough.

The resolution / quality of the Xenium plots in 6i is low, same for 6j left.

The authors mention that Xenium confirms features at subcellular level. Based on the revised manuscript, it is unclear to me which features the authors refer to; I could not find any analyses that take into account the subcellular transcript distribution. I would clarify or omit subcellular.

Reviewer #3 (Remarks to the Author):

The authors have addressed all my comments and concerns.

Thank you.

I do not have other additional comments.

Point-by-point responses to Reviewer #2:

Reviewer #2 (Remarks to the Author):

The authors have revised their scientific manuscript, providing thorough responses to each of the reviewer's comments. The authors addressed my concerns regarding data availability by making the raw sequencing data publicly available in the Japanese Genotype-phenotype Archive of the National Bioscience Database Center (NBDC/JGA), which they acknowledge has a restricted access due to ethical guidelines in Japan. Furthermore, they have deposited the processed data, such as lists of somatic mutations, DNA methylation profiles, and gene expression levels, in an open-access database. Regarding the computational analyses reproducibility, the authors have made their scripts publicly available on GitHub, allowing for the replication of their analyses.

In response to the reviewer's comment on potential confounding variables in bulk RNA samples, the authors used deconvolution analyses of bulk RNA-seq and Visium data. They compared these to single-cell RNA-seq data of lung adenocarcinoma, which identified various infiltrating immune cell types and subtypes of macrophages. They further validated these findings at a single-cell level using Xenium data.

I very much appreciate the addition of the Xenium dataset to this revised manuscript. It strengthens the conclusions regarding the TME composition. However, the current state of analysis seems a bit premature in the revised manuscript.

We appreciate the reviewer's suggestions. We performed the indicated Xenium analysis for the Noguchi type A specimen (case TSU-20). We also conducted additional extensive analyses of Xenium data to strengthen of the conclusions about local macrophage patterns.

Specific comments:

2-1: The manuscript compares gene expression features between Noguchi type A and B samples. The analysis with the custom xenium lung panel was only performed with one Noguchi type B sample. Could the authors also analyze a Noguchi type A sample for

comparative reasons?

We thank the reviewer for this careful comment. According to the reviewer's suggestion, we conducted the indicated Xenium analysis for the AIS Noguchi type A specimen (**Supplementary Fig. S12** and **Supplementary Table S10**) which we have used the Visium analysis in the original manuscript. As partly expected, the Xenium analysis showed that there are a number of variable normal cells, such as endothelial cells and fibroblasts, indicating that the normal lung structures are still retained (**Supplementary Fig. S12b**). As for the cancer cells, their gene expression pattern was rather uniform, including *SFTPB* and *NAPSA*-high well-differentiated cells (**Supplementary Fig. S12c**). As for the immune cells, although there are some inflammatory cells already recruited, reflecting the cancer cells are already resided along with the alveoli structures, no relevant sign of the malignant-transformation was observed. For example, a number of macrophage cells were observed, but most of them are not SPP1+ macrophages (**Supplementary Fig. S12e**). This is a sharp contrast to the case of Noguchi type B, where far larger number of SPP1+ macrophages were observed with spatial proximity with FABP4+ macrophages (also see the below **Comment 2-4**). Those observations supported our notion that, at the stage of Noguchi type A, the drastic status changes of the cancer cells and their intensive interaction of the stromal cells has not been initiated. We included this analysis to the revised manuscript (**lines 392–400**).

2-2. I appreciate that the authors list the obtained transcript counts per cell in 6i. Could the authors comment in the manuscript or methods and contextualize this capture rate / the corresponding RNA quality of the sample and what one would expect based on the information supplied by the manufacturer or the literature?

We thank the reviewer for this constructive comment. We confirmed that the number of detected transcripts is compatible with that in the report by 10x Genomics. According to the reviewer's suggestion, we included the citation of the literature (*) and mentioned the quality control in the **Methods** section (**lines 592–594**). We also added information on the negative control rates in **Supplementary Table S10** and confirmed that these values were sufficiently low.

*Jenesick A et al. High resolution mapping of the breast cancer tumor microenvironment using integrated single cell, spatial and in situ analysis of FFPE tissue. 2022 bioRxiv doi: 10.1101/2022.10.06.510405

2-3. The TME composition in 6j is illustrated with a few marker genes of interest, could this analysis also be performed with the detected clusters in 6i (unsupervised)?

Of course, it is possible to conduct the similar analysis for other context of the clusters appearing in **Figure 6i**. We added the figure of the detected clusters in **Supplementary Figure S10g** and confirmed that a number of cells which were assigned to the macrophage cluster (cluster 1) and SPP1+ macrophage cluster (cluster 9) were resided in this region.

2-4. The authors mention macrophage localization patterns in 6j, it would be nice to have quantitative measures of this localization pattern and statistical test (neighborhood analysis, distance..) that compare it with a control sample. The current phrasing “FABP4+ and SPP1+ macrophages were localized in certain alveoli structures” is not specific enough.

We conducted the indicated quantitative analysis. The results are shown in **Supplementary Figure S11**. Precisely, we counted the number of FABP4+ and SPP1+ macrophages and their mutual distance. We found that their proximity was relevant (showing up in red) at the region where the immune suppressive reaction of the cancer is starting (Areas 54-56 in **Supplementary Figure S11**). The comparison with Noguchi type A is also shown in **Supplementary Figure S12e**.

2-5. The resolution / quality of the Xenium plots in 6i is low, same for 6j left.

We updated the Xenium plots in **Figures 6i** and **6j** left.

2-6. The authors mention that Xenium confirms features at subcellular level. Based on the revised manuscript, it is unclear to me which features the authors refer to; I could not find any analyses that take into account the subcellular transcript distribution. I would clarify or omit subcellular.

We are sorry for the confusion. As indicated by the reviewer, the sub-cellular level resolution analysis is not an issue of this study. We changed “sub-cellular” into “single-cell” in the revised text (**lines 396 and 576**).

REVIEWERS' COMMENTS

Reviewer #2 (Remarks to the Author):

I have no further comments and recommend publication of the manuscript.